# Miniature computational spectrometer with a plasmonic nanoparticles-in-cavity microfilter array

Yangxi Zhang [1,4], Sheng Zhang [2,4], Hao Wu [1], Jinhui Wang [1], Guang Lin [2,3] ✉ & A. Ping Zhang [1] ✉

Optical spectrometers are essential tools for analysing light–matter interactions, but conventional spectrometers can be complicated and bulky. Recently, efforts have been made to develop miniaturized spectrometers. However, it is challenging to overcome the trade-off between miniaturizing size and retaining performance. Here, we present a complementary metal oxide semiconductor image sensor-based miniature computational spectrometer using a plasmonic nanoparticles-in-cavity microfilter array. Size-controlled silver nanoparticles are directly printed into cavity-length-varying Fabry–Pérot microcavities, which leverage strong coupling between the localized surface plasmon resonance of the silver nanoparticles and the Fabry–Pérot microcavity to regulate the transmission spectra and realize large-scale arrayed spectrum-disparate microfilters. Supported by a machine learning-based training process, the miniature computational spectrometer uses artificial intelligence and was demonstrated to measure visible-light spectra at subnanometre resolution. The high scalability of the technological approaches shown here may facilitate the development of high-performance miniature optical spectrometers for extensive applications.

The optical spectrum carries essential information of a light beam; various optical spectroscopy technologies have been developed as powerful characterization and measurement tools, such for studying the structures of atoms and molecules, analyzing material compositions, and remotely sensing distant targets such as satellite imaging and astronomical observation[1–5]. Traditional spectrometers are bulky and highly sophisticated instruments that are only suited for laboratory-based use. Dispersive elements, such as diffraction gratings, are typically used to discriminate light components of different wavelengths for spectral measurement and analysis[1,2]. These spectrometers are bulky because of the use of relatively large sizes of these dispersive components and their corresponding need for mechanical movement or rotation for resolving wavelengths. Alternatively, tunable optical filters or scanning-form interferometers may be used[6,7].

For instance, Fourier transform spectrometers (FTSs) based on Michelson interferometers are widely utilized in infrared spectroscopy testing and measurement[8,9]. However, the use of a Michelson interferometer with a movable mirror leads to bulky and long response times.

Recently, to unlock new spectroscopy applications for on-site measurement and portable tools, many efforts have been invested in developing low-cost, small, and easy-to-use computational spectrometers[1,2]. In computational spectrometers, incident light is sampled by a wavelength-selective component, and the sampled light spectrum is reconstructed from the measured optical response of photosensors. Such computational spectrometers fully leverage the high computing power of microprocessors. With sophisticated spectrum reconstruction algorithms, the spectral resolving ability of

[1]Photonics Research Institute, Department of Electrical and Electronic Engineering, The Hong Kong Polytechnic University, Kowloon, Hong Kong SAR, China. [2]Department of Mathematics, Purdue University, West Lafayette, IN, USA. [3]School of Mechanical Engineering, Purdue University, West Lafayette, IN, USA. [4]These authors contributed equally: Yangxi Zhang, Sheng Zhang. ✉e-mail: guanglin@purdue.edu; azhang@polyu.edu.hk

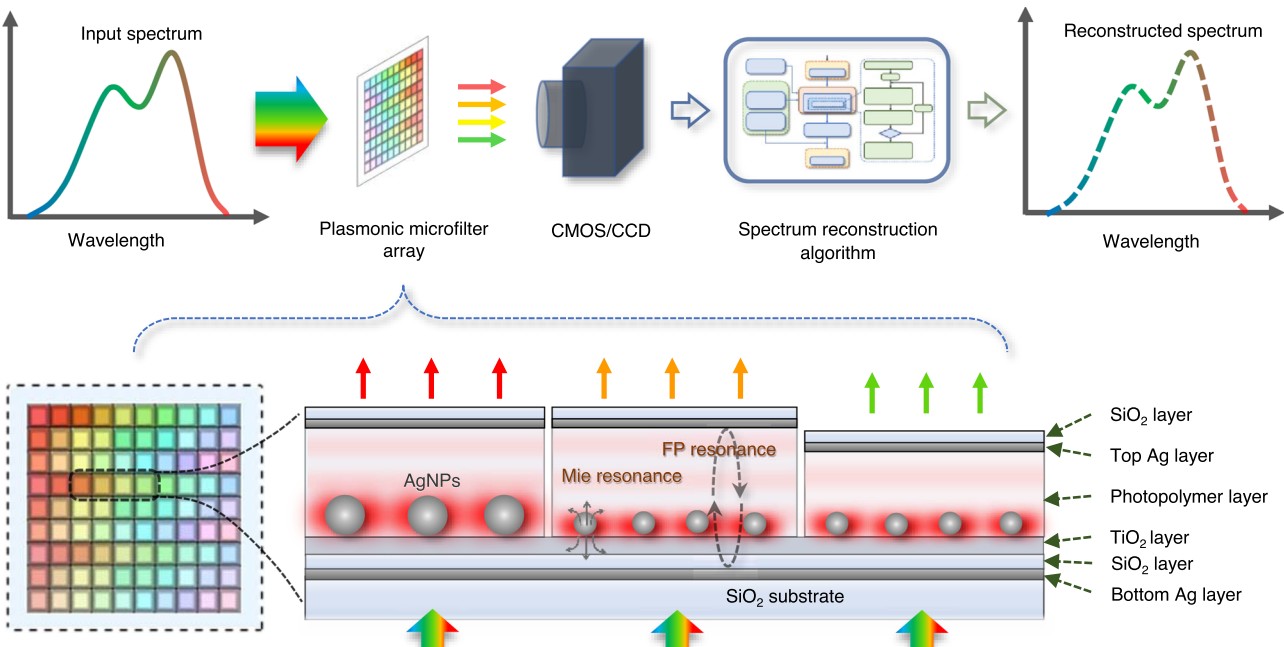

**Fig. 1 | Schematics of the working principle of a miniature computational spectrometer and its plasmonic nanoparticles-in-cavity microfilters.** The top section illustrates the working principle of our proposed miniature computational spectrometer, while the bottom section shows the cross-section of the plasmonic microfilter array, in which both the size of AgNPs and the length of FP cavities are designed to create a large-scale spectrum-disparate microfilter array. A more detailed diagram of the spectrum reconstruction algorithm is shown in the Supplementary Information.

spectrometers can be greatly improved, e.g., by enabling a commercial RGB image sensor with only three broadband color filters for effective spectrum measurement[10,11] and decreasing the orthogonal requirement of wavelength-selective components in spectrometer construction. Therefore, many types of photonic technologies, such as thin-film optical filters[12], perovskite film[13], single nanowire and superconducting nanowire[14,15], tunable van der Waals junction[16], folded digital metalenses[17], integrated photonic chips[18,19], and wavelength-selective photodetector[13,20], have been used as computational spectrometers for integrated chip-size spectrometers.

In particular, complementary metal oxide semiconductor (CMOS) or charge-coupled device (CCD) image sensor-based miniature spectrometers have been widely considered the most viable candidates for resolving the tradeoff between miniaturizing size and retaining performance. They offer the capability of integrating millions (and even tens of millions) of photosensors while being very compact and robust; additionally, they have been widely tested in smartphones and lab-on-a-chip systems. Similar to how encoding capacity can be increased by the length of coding, the resolution of such CMOS/CCD image sensor-based computational spectrometers can be efficiently enhanced by increasing the number of spectrum-disparate filters. Therefore, developing miniature spectrometers by devising optical microfilter arrays with many spectrum-disparate elements and spectrum reconstruction algorithms has recently come to the forefront. Many types of microfilters, such as quantum dots[21,22], photonic crystals[23–26], plasmonic encoders[27], plasmonic rainbow chips[28], metamaterials[6,11,15,29,30], liquid crystals[31], random structures[32,33], multilayer film filters and interference filters[12,34–38], and 3D-printed microoptics[39], have been demonstrated to be useful for computational spectrometry. Multilayer film filters have good flexibility in tailoring optical spectral responses[12,36,37]. However, their fabrication processes using many steps of deposition/patterning are less scalable in the fabrication of large-scale distinct filter arrays. Nanophotonic methods that realize the optical responses of nanostructures via material and geometric nanoengineering have become among the most promising approaches. For instance, plasmonic 2D chirped gratings, in which the

rainbow trapping effect is utilized to split wavelengths[40], have been used to construct a dual-function computational spectrometer for simultaneous spectroscopic and polarimetric analysis. However, the optical transmittance of such metallic 2D chirped gratings is low; thus, computational spectrometers can operate in reflection mode only, which may limit further miniaturization for portable applications. Metamaterial/metasurface structures have also been used to fabricate optical microfilter arrays for computational spectrometers[6,11], and very promising results with subnanometer spectral resolution have been demonstrated[29,30]. However, such nanophotonic filters usually require the use of either e-beam lithography (EBL) or nanoimprinting with EBL-fabricated nanotemplates to fabricate many ultrafine structures at the nanometer scale. These nanofabrication processes are very time-consuming and expensive and thus limit the total number of manufacturable spectrum-disparate microfilters. Therefore, innovative designs are still needed to alleviate the dependence of nanophotonic microfilters on nanometer-scale geometry to enable the fabrication of very large-scale arrayed spectrum-disparate microfilters for high-performance computational spectrometers.

In this paper, we present an artificial intelligence (AI)-empowered miniature spectrometer based on a highly scalable plasmonic nanoparticles-in-cavity microfilter array to resolve the above-mentioned challenges. As shown in Fig. 1, the plasmonic microfilter array is used to sample incident light before the light beam is detected by a digital image sensor. The image data carry spectral information about the incident light and can be used to reconstruct the spectrum of incident light via algorithms. The plasmonic nanoparticles-in-cavity structure, in which both the length of the Fabry–Pérot (FP) interferometric microcavity and the size of the silver nanoparticles (AgNPs) can be precisely tailored, is designed to harness the transmission spectra of the microfilters. Owing to the strong coupling between the FP resonance and the localized surface plasmon resonance or Mie resonance of AgNPs, such a structure greatly enhances the spectral diversity to develop large-scale arrayed optical disparate microfilters. Moreover, a digital ultraviolet exposure-based fabrication technology was established to directly print both size-controlled AgNPs and

length-varying FP cavities to ensure the high scalability of the plasmonic microfilter array. Leveraging a machine learning-based training process, the AI-empowered miniature spectrometer with a plasmonic microfilter array of more than a thousand spectrum-disparate microfilters was demonstrated to measure visible-light spectra at subnanometer resolution.

## Results

### Plasmonic nanoparticles-in-cavity microfilter array

Microcavity plasmonics with strong coupling between optical microcavities and plasmons have recently emerged as tools for tailoring the optical responses of photonic structures[41,42]. Here, we incorporate AgNPs into an FP microcavity to harness the strong coupling between FP resonance and Mie resonance for the creation of large-scale arrayed microfilters with different transmission spectra. When considering a layer of random and dense AgNPs, diffraction orders resulting from a periodic lattice of scatterers can be neglected[43,44]. The corresponding reflection and transmission can be expressed as functions of frequency, which is known as Mie resonance. For small AgNPs, the reflection from such an AgNP layer is weak. When AgNPs are located at one of the antinodes of the FP mode, the strong coupling in the cavity leads to Rabi splitting, which can be expressed as[41,43,44]:

$$\frac{1}{\lambda_\pm} = \frac{1}{\lambda_0}\left(1 \pm \sqrt{\frac{2\pi V}{AL}}\right) \quad (1)$$

where $\lambda_0$ is the wavelength matched between the plasmonic resonance and the FP cavity mode, $\lambda_+$ and $\lambda_-$ are the split resonant peak positions, $V$ is the effective scatter volume, $A$ is the unit cell area, and $L$ is the equivalent length of the FP cavity. This reveals that a high-density metal nanoparticle layer is favorable for large Rabi splitting, and the splitting magnitude is inversely proportional to the square root of the cavity length. When the AgNP layer is at the center of the cavity, odd modes of the FP cavity interact with the plasmons of the Mie resonance; when the AgNP layer is at other positions, even modes may also interact with plasmons. For longer etalon lengths $L$ and high-order modes, the splitting effect is reduced. Notably, as the concomitant on-resonance quasistatic polarizability of Mie resonance can far exceed the polarizability attained by a single dipole scatter[41,45], a low-$Q$ Mie resonance of AgNPs can also lead to strong coupling.

Considering such coupling mechanisms, we design a plasmonic nanoparticles-in-cavity microfilter to make large-scale arrayed thin-film microfilters with different transmission spectra, as shown in Fig. 1. In this design, a photopolymer layer together with two silver nanolayers is used to form FP microcavities; therefore, the length of FP microcavities can be precisely tailored by dynamic UV exposure technology. Moreover, with the incorporation of a titanium dioxide photocatalytic nanolayer, these size-controlled AgNPs can be directly printed by precision photoreduction technology[46,47], avoiding the use of expensive nanofabrication processes such as EBL. In contrast to microfilters engineered by either FP microcavity length or AgNP size alone, combinations of $n$ FP microcavity lengths and $m$ AgNP sizes can provide $n \times m$ different microfilters and thus endow such thin-film microfilter devices with high scalability for high-resolution broadband computational spectrometers.

Figure 2 shows the simulation results of the transmission spectra of these plasmonic nanoparticles-in-cavity microfilters. The thickness of the two silver mirror layers is optimized to 25 nm to balance the transmission and the resonance strength of the FP cavity. The thicknesses of the $SiO_2$ protective layer and $TiO_2$ photocatalytic layer, which form the space between the AgNPs and the bottom Ag mirror, are set to 43.5 nm and 10 nm, respectively. Figure 2a(i–iii) shows the transmission spectra of the plasmonic microfilters when the thickness of the photopolymer layer is 250, 320, or 400 nm. In the 250 nm case (see Fig. 2a(i)), when the AgNP diameter increases, a redshift of the 2nd FP

peak can be observed, even though the 2nd FP peak (500–600 nm) is not exactly at the position of the plasmonic resonance mode (~400 nm). In the 320 nm case (see Fig. 2a(ii)), when the AgNP diameter increases, a typical Rabi splitting of the 3rd FP peak can be observed at wavelengths ranging from 400 to 450 nm, as this coincides with the plasmonic resonance mode. A redshift of the 2nd FP peak at 600–700 nm can also be observed. In the 400 nm case (see Fig. 2a(iii)), when the AgNP diameter increases, the 2nd FP peak and the 3rd FP peak both redshift, while the 4th FP peak blueshifts. These results show that the introduction of AgNPs directly causes Rabi splitting and also causes a Rabi splitting-like opposite shift to FP modes on two sides of the plasmonic resonance mode, even if those sides are far from the plasmonic resonance frequency near 400 nm, such that the response can effectively cover the visible spectrum. Notably, these shifts are not caused by the unidirectional equivalent cavity length change but rather by the coupling of the two resonant systems, thus resulting in opposite shift directions.

Figure 2b(i, ii) shows the transmission spectra of the plasmonic microfilters when the average diameters of the AgNPs are 10 nm and 8 nm, respectively, while the thickness of the photopolymer layer changes from 260 to 350 nm. With the movement of the FP peaks according to the change in cavity length, another typical phenomenon of strong coupling, anticrossing, can be observed in the movement of the Rabi splitting peaks (i.e., the FP⁻ peak with its peak wavelength $\lambda^-$ and the FP⁺ peak with its peak wavelength $\lambda^+$), as shown in Fig. 2c.

More detailed numerical simulation results are provided in Fig. 2d, e. Figure 2d shows the transmission spectrum of the total microfilter structure when the diameter of the AgNPs changes from 0 to 60 nm, and the thickness of the photopolymer is fixed at 200, 300, and 400 nm. Figure 2e shows the transmission spectrum of the total microfilter structure when the thickness of the photopolymer layer changes from 50 to 500 nm, and the diameter of the AgNPs is fixed at 0, 10, or 20 nm. These results demonstrate that changing the thickness of the photopolymer layer (the length of the FP cavity) and the size of the AgNPs, both exhibit clear modulatory effects on the overall spectral response of the strong coupling structure, which can realize a spectral variety of filters.

### Fabrication and characterization of the plasmonic microfilter array

The fabrication of such a plasmonic microfilter array includes five steps: sputtering deposition of silver and $SiO_2$ nanolayers (acting as a bottom mirror and protective layer, respectively); sol-gel spin coating of the $TiO_2$ nanolayer; direct printing of size-controlled AgNPs; grayscale patterning and nanoscale thickness tuning of polymer FP cavities; and sputtering deposition of silver and $SiO_2$ nanolayers (acting as a top mirror and protective layer, respectively), as shown in Fig. 3a. Two critical steps of the fabrication process are direct printing of size-controlled AgNPs and nanoscale thickness tuning of the polymer FP cavity, which are the keys to achieving a large-scale spectrum-disparate microfilter array.

Recently, we developed a precision photoreduction technology to additively print size-controlled AgNPs or gold nanoparticles (AuNPs) using a digital UV exposure technique for plasmonic applications[46,47]. With the use of an opaque substrate with a silver mirror layer (for FP microcavities), however, previous printing processes cannot be directly applied, as the UV light must illuminate the $TiO_2$ photocatalytic layer through the silver salt solution instead of through the bottom $SiO_2$ substrate side. There were two challenges associated with this technique: (i) UV light may directly trigger unwanted photoreduction reactions in silver salt solution; (ii) the scattering of UV light may increase with the growth of AgNPs, and the resulting UV pattern may become too blurry to print a well-defined micropattern of AgNPs. To overcome these challenges, we modified the previous material formulation. Ethylene glycol was used to replace

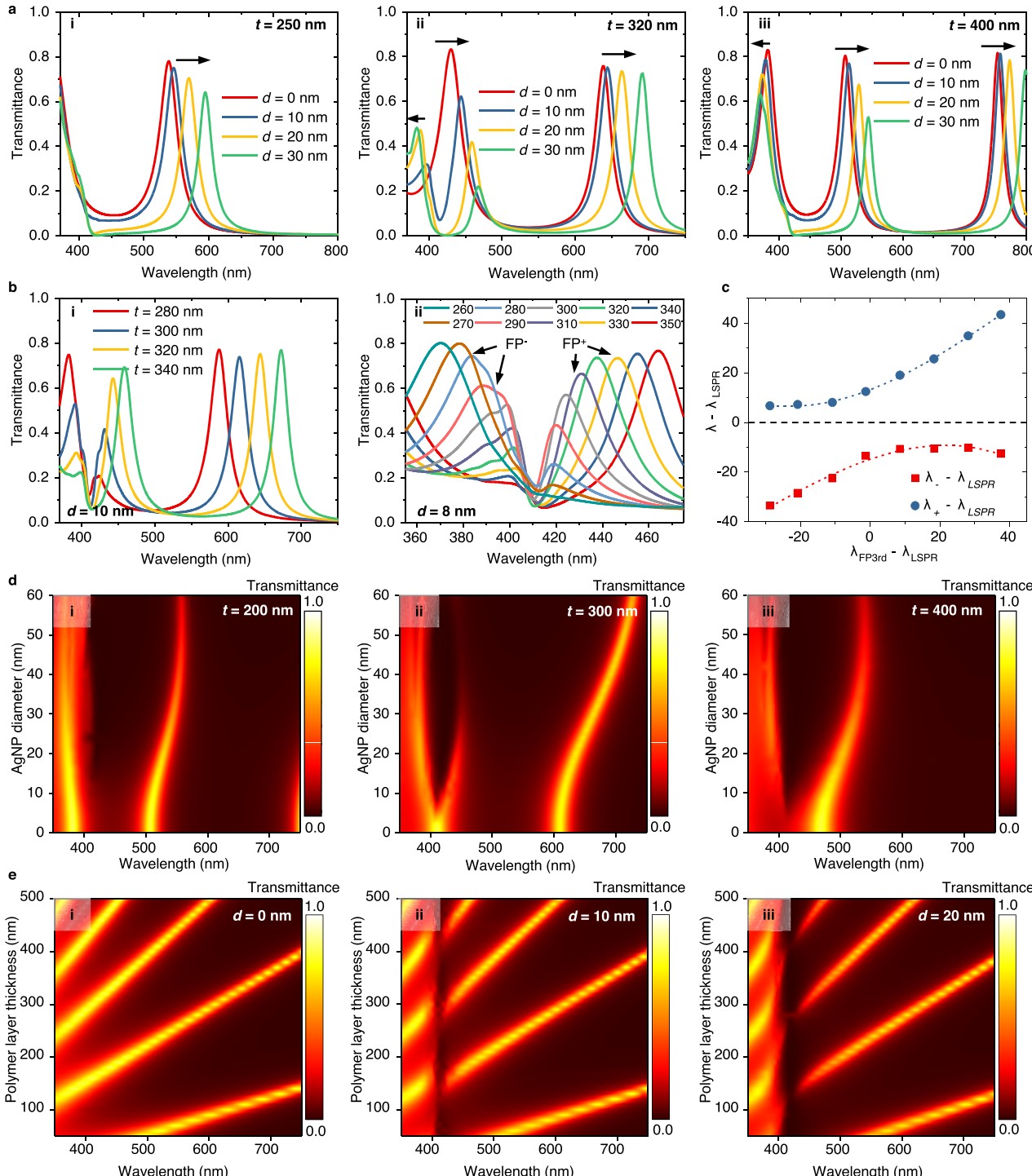

**Fig. 2 | Numerical simulation results of plasmonic nanoparticles-in-cavity spectrum-disparate microfilters. a** Rabi splitting and opposing wavelength shift phenomena: (i–iii) the transmission spectra of the microfilters with different sizes of AgNPs when the thickness of the photopolymer layer(t) is 250, 320, and 400 nm, respectively. **b** Wavelength shifts caused by the change of photopolymer layer thickness: (i, ii) the transmission spectra of the microfilters with different thicknesses of the photopolymer layer when the diameter of AgNPs(d) is 10 and 8 nm, respectively. **c** Anti-crossing phenomenon, based on the peak wavelength data from (**b**(ii)). **d** Modulation effect of AgNP size on the transmission spectra of total microfilter structure: (i–iii) the transmission spectra of the microfilters whose photopolymer layer has the thickness of 200, 300, and 400 nm, respectively. **e** Modulation effect of polymer layer thickness on the transmission spectra of total microfilter structure: (i–iii) the transmission spectra of the microfilters with 0, 10, and 20 nm AgNP diameters, respectively.

water and glucose to act as both a solvent and reducing agent. The previously used additive PVP was removed from the solution. Experiments revealed that such an adjustment can slightly lower the activity of the silver salt solution and thus enhance the stability of the printing process. Figure 3b shows SEM images of the printed AgNPs, from

which one can see that AgNPs of different sizes with diameters ranging from 0 to 50 nm are printed through the control of the UV exposure dose.

To rapidly fabricate polymer cavity length-varying FP microcavities, a digital grayscale photopolymerization technology, as shown

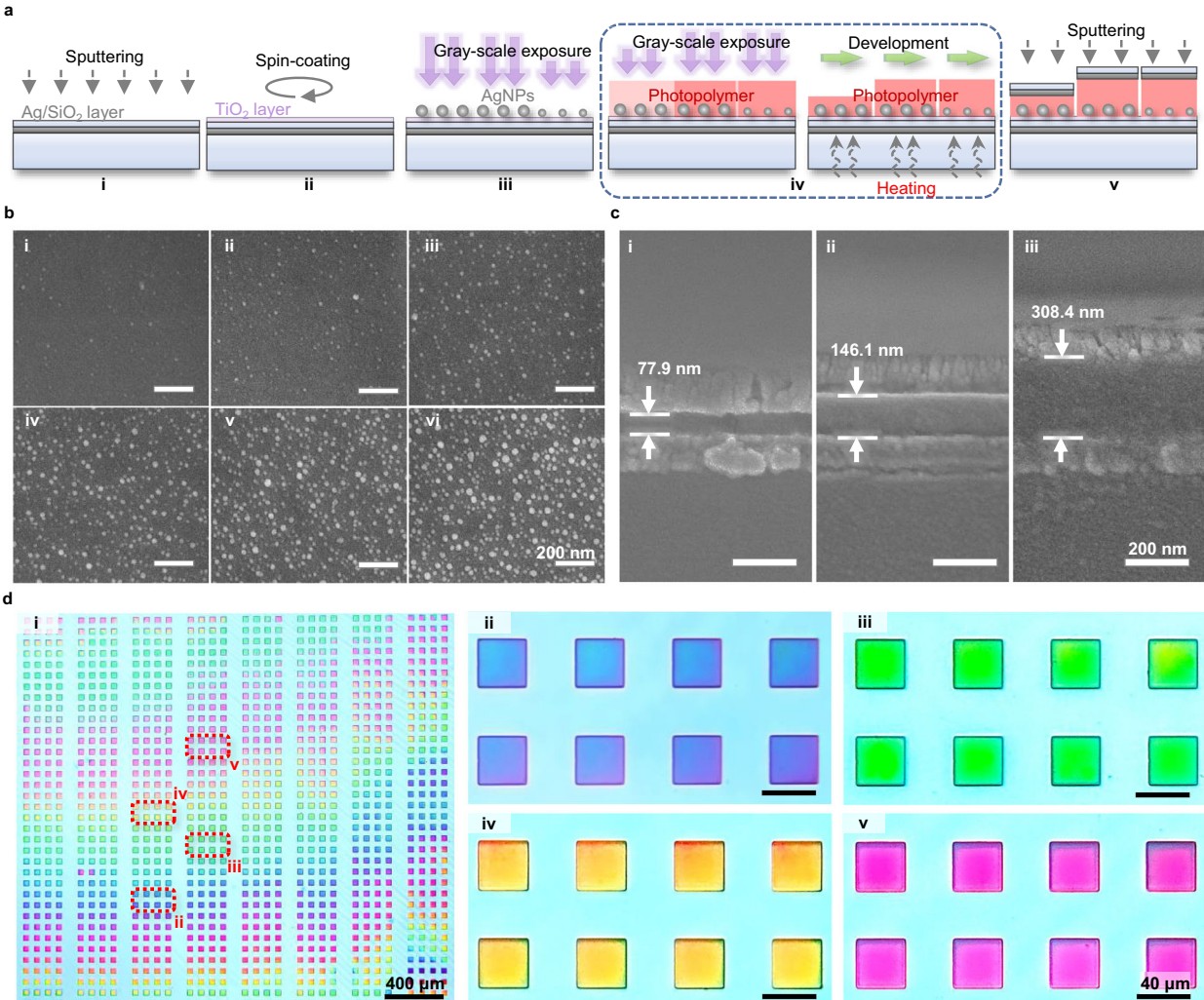

**Fig. 3 | Fabrication of plasmonic nanoparticles-in-cavity microfilter array.**
**a** Schematics of fabrication processes: (i) sputtering of Ag/SiO₂ layers; (ii) spin coating of TiO₂ nanolayer; (iii) direct printing of size-controlled AgNPs; (iv) grayscale patterning of polymer FP cavities; (v) sputtering of Ag/SiO₂ layers. **b** SEM images of printed AgNPs. **c** SEM images of the cross-sections of the fabricated polymer thickness-tuned FP microcavities. **d** Transmissive optical images of the fabricated plasmonic microfilter array: (i) the whole sample image with 1152 microfilters; (ii-v) partially enlarged images of eight microfilters.

in Fig. S1, was developed to enable rapid micrometer-scale patterning and ultrafine nanometer-scale thickness tuning of photopolymer films. More specifically, an oxygen inhibition mechanism of free-radical polymerization is used in combination with grayscale optical exposure to tailor the polymerization of the polymer in the vertical direction. Oxygen inhibition leads to an induction period and can significantly slow polymerization[48–50]. For the dipentaerythritol hexaacrylate (DPHA) film with a thickness of hundred nanometers used in our experiments, the diffusion of oxygen from the air to the film results in a gradient of oxygen inhibition in the vertical direction. UV light can be exponentially absorbed in the same direction. Consequently, polymerization may initiate from the bottom and extend to the top of the photopolymer film, and the degree of polymerization in the upper section will be relatively lower than that in the bottom section. In the following development process, oligomers with a low degree of polymerization may be dissolved by the solvent. After development, the polymer film is further baked at a relatively high temperature (at 120 °C for 2 h) to evaporate the solvent and achieve a stable nanofilm. The reactivity and viscosity of the monomer, the concentration of the photoinitiator, and the intensity of the UV light were optimized to flexibly tune the thickness of the micropatterned polymer at the nanoscale. Typical cross-sectional SEM images of the fabricated photopolymer nanofilms with different thicknesses are shown in Fig. 3c.

Notably, both fabrication techniques are based on digital ultraviolet exposure technology using a digital micromirror array (DMD). A high-speed DMD has a switching time of less than one millisecond, which enables ultrafast, parallel modulation of UV exposure doses at different target positions. The size of the AgNPs and the thickness of the FP microcavities can be rapidly and finely tailored via precision photoreduction and grayscale photopolymerization, respectively, to fabricate the designed large-scale disparate microfilter array. Figure 3d shows a fabricated sample with a total of 1152 microfilters. The size of each microfilter element is 38.1 μm × 38.1 μm, while the total size of the microfilter array is 2.914 mm × 2.690 mm. The color of the transmissive optical microscope images of the fabricated microfilters periodically change from blue to red according to the tuning of the fabrication parameters, which reveals the great flexibility of such precision photoreduction and digital greyscale photopolymerization combined fabrication processes. Additional intermediate results during the production process can be found in Fig. S2a, b, and additional fabricated microfilter samples can be seen in Fig. S2c.

The measured transmission optical spectra of the fabricated large-scale arrayed plasmonic microfilters are shown in Fig. 4. Figure 4a shows that such a plasmonic microfilter array provides high-density transmission spectral peaks, i.e., 2436 transmission peaks, ranging from

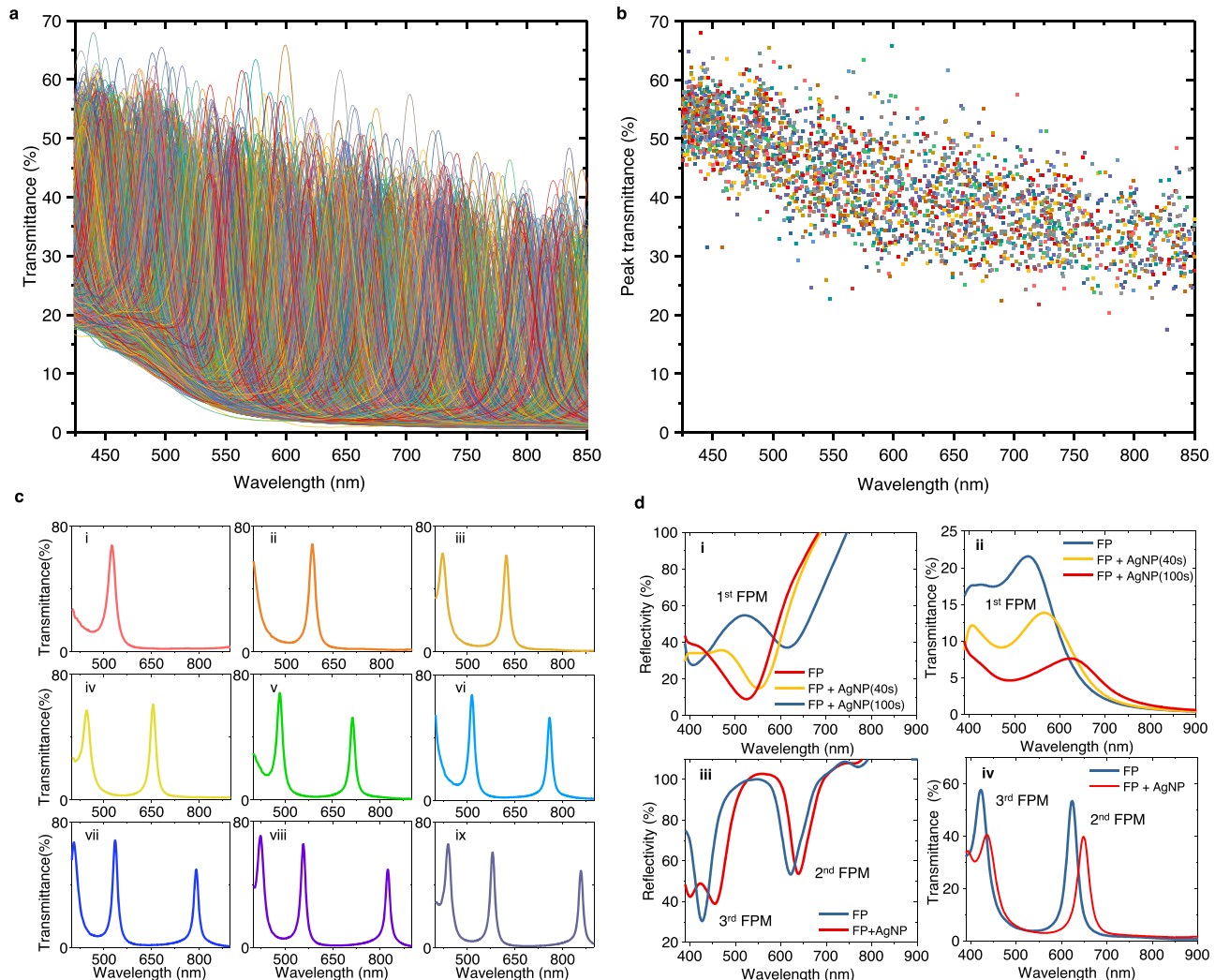

**Fig. 4 | Measured spectra of the fabricated plasmonic nanoparticles-in-cavity microfilters. a** Transmission spectra of all elements of the fabricated microfilter array. **b** Distribution of the peak wavelengths of all elements of the fabricated microfilter array. **c** Typical transmission spectra of microfilter elements: i-ix, the transmission spectra of the microfilters whose photopolymer layer has the thickness of ~240, ~280, ~310, ~330, ~370, ~410, ~430, ~450, and ~480 nm,

respectively. **d** Typical Rabi splitting and peak wavelength shifts in reflection and transmission spectra: (i, ii) the reflection and transmission spectra of a microfilter whose photopolymer layer has a thickness of ~60 nm; (iii, iv) the reflection and transmission spectra of a microfilter whose photopolymer layer has a thickness of ~310 nm.

425 to 850 nm. The full widths at half-maximum (FWHMs) of those transmission spectral peaks range from 12.2 to 88.3 nm, with an average value of 23.8 nm. The wavelength and transmittance distributions of these spectral peaks are shown in Fig. 4b. Typically, measured transmission spectra are given in Fig. 4c, which shows that these microfilters have sharp transmission peaks varying in the wavelength range from 400 to 900 nm. The peak density distributions and estimated ideal spectral resolving abilities of the fabricated plasmonic microfilters are shown in Fig. S3.

Moreover, Rabi splitting, which was predicted via numerical simulation, was observed in the measured reflection and transmission spectra in the short wavelength region. When the cavity length is short, a broad resonance peak of the 1st-order FP mode appears in the wavelength range of visible light. With the increasing size of the AgNPs, clear spectral splitting was observed at approximately 400–600 nm in both the reflection and transmission spectra, as shown in Fig. 4d(i, ii). However, the depth of the transmission dip caused by Rabi splitting is not as deep as that in the numerical simulation results, which may be attributed to the variation in the size of the printed AgNPs. When the cavity length becomes

relatively longer, the resonance spectral peaks of the 2nd- and 3rd-order FP modes enter the visible light range (see Fig. 4d(iii) and 4-d(iv)). For this case, the resonance spectral peak of the 3rd FP mode shows Rabi splitting because its resonance wavelength is close to that of plasmonic AgNPs, while the resonance spectral peak of the 2nd FP mode remains in a longer wavelength region and may shift to a longer wavelength with increasing size.

The polarization dependence of these plasmonic microfilters was measured with a polarization-switchable input light beam, as shown in Fig. S4. The transmittance of plasmonic microfilters changes minimally when the polarization of the linearly polarized input light varies from 0 to 180°. The measured polarization dependence is approximately 0.086 dB, which indicates that the computational spectrometer is not sensitive to the polarization of the input light. The cross-correlation coefficients and their statistical distributions in the transmission spectra of the fabricated plasmonic microfilters are presented in Fig. S5. The majority (~70.6%) of the cross-correlation coefficients are between 0.2 and 0.6. This result signifies the necessity of the use of the AI method for spectrum reconstruction in our computational spectrometer.

## AI-empowered miniature computational spectrometer

A machine-learning-based AI method was established to enable the spectrometer to use a large-scale plasmonic microfilter array to reconstruct the spectrum of an input light beam from the CMOS image output. Suppose that there are $m$ microfilters and $n$ training data points that are the pairs of input light spectra and CMOS outputs: $(\mathbf{x}_j, \mathbf{y}_j)$, $j = 1, ..., n$. An input light spectrum $\mathbf{x}_j(\lambda)$ is a function that maps each wavelength $\lambda$ to the intensity of the light at that wavelength. When a microfilter array of $m$ elements is used, we have a CMOS image output with $m$ effective pixel values $\mathbf{y}_j = [y_{j,1}, ..., y_{j,m}]^T$ for each input light spectrum $\mathbf{x}_j$. For a new unknown input light spectrum $\widetilde{\mathbf{x}}(\lambda)$, the problem is how to reconstruct $\widetilde{\mathbf{x}}(\lambda)$ from the CMOS output $\tilde{\mathbf{y}} = [\tilde{y}_1, ..., \tilde{y}_m]^T$. To achieve efficient spectrum reconstruction, we assume that i) the training data, i.e., the input single narrow-peak light spectra, are sufficient and ii) the reconstruction target is a continuous spectrum, which facilitates the use of denoising methods such as LASSO, ridge regression, total variation (TV), and quadratic variation (QV).

To ensure the effectiveness of the training process, we apply a series of single-narrow-peak light spectra generated by a tunable monochromator as training data. Then, any input light spectrum $\widetilde{\mathbf{x}}$ can be effectively constructed by a linear combination of such training light spectra $\mathbf{x}_j$ as:

$$\tilde{x}(\lambda_i) \cong \sum_{j=1}^{n} w_j x_j(\lambda_i), \qquad (2)$$

where $\lambda_i$ are the discretized wavelengths under monitoring by a calibration spectrometer. Because the CMOS sensor typically has a linear response to input light, i.e., $\mathbf{y}$ linearly depends on $\mathbf{x}$, the relationship between the CMOS outputs $\tilde{y}$ and $\mathbf{y}_j$ can be derived from Eq. (2) as:

$$\tilde{y} \approx \sum_{j=1}^{n} w_j y_j. \qquad (3)$$

Equivalently

$$\begin{bmatrix} \tilde{y}_1 \\ \vdots \\ \tilde{y}_m \end{bmatrix} \approx \begin{bmatrix} y_{1,1} & \cdots & y_{n,1} \\ \vdots & \ddots & \vdots \\ y_{1,m} & \cdots & y_{n,m} \end{bmatrix} \begin{bmatrix} w_1 \\ \vdots \\ w_n \end{bmatrix} \qquad (4)$$

Therefore, the problem of finding $w_j$, where $j = 1, ..., n$, for the spectrum construction is transformed into solving the following optimization problem:

$$\begin{aligned} Minimize\, Loss\,(w_1, \cdots, w_n) = &\sum_{k=1}^{m} \left( \tilde{y}_k - \sum_{j=1}^{n} w_j y_{j,k} \right)^2 + c_1 \sum_{j=1}^{n} |w_j| + c_2 \sum_{j=1}^{n} w_j^2 \\ &+ \sum_{p=1}^{n-1} c_{3,p} \sum_{j=p+1}^{n} |w_j - w_{j-p}| + \sum_{q=1}^{n-1} c_{4,q} \sum_{j=q+1}^{n} (w_j - w_{j-q})^2, \end{aligned}$$
$$(5)$$

where $c_1$, $c_2$, $c_{3,p}$, and $c_{4,q}$ are nonnegative hyperparameters. The first term is a least-squares term used to minimize the discrepancy between the calculated CMOS response of the reconstructed spectrum and the measured CMOS readings. The last four terms are regularization terms used to denoise and ensure the robustness of the machine learning algorithm.

The hyperparameter $c_1$ influences the $L_1$ norm of the parameter vector $\mathbf{w}$ and induces its sparsity such that $\mathbf{w}$ has fewer nonzero components. This hyperparameter is also known as the least absolute shrinkage and selection operator (LASSO)[51]. In the context of our spectrum reconstruction, a larger $c_1$ promotes sparser reconstruction (for example, low intensities are filtered out as noise, resulting in a narrow-peak light spectrum with zero intensity at most wavelengths).

The hyperparameter $c_2$ is used for $L_2$ norm regularization of the parameter vector $\mathbf{w}$. This hyperparameter is also used in ridge regression[52]. $L_2$ norm regularization reduces noise and adds stability to the algorithm when the training data are highly correlated. When large-scale arrayed microfilters with weak spectral orthogonality are used in our computational spectrometer, the training process produces many similar output images. Numerically, this trend causes the least-square regression problem in Eq. (4) to be ill-posed, resulting in non-identifiability and overfitting. A corresponding solution is to use $L_2$ norm regularization by setting $c_2 > 0$, which restricts large components of the parameter vector $\mathbf{w}$ and thus has a smoothing effect on the solution. From the optimization perspective, the quadratic penalty term in the loss function (Eq. (5)) results in a strongly convex function[53]; e.g., the matrix $\nabla^2 Loss - 2c_2 I$ is positive semidefinite for all elements except a subset of zero measurements, which endows the algorithm with many advantages, such as (i) a unique global minimum without other local minima and (ii) fast convergence.

The hyperparameter $c_{3,1}$ leads to the term TV, which often appears in fused LASSO[22,54,55] to enhance the local constancy of the coefficient profile. In our context, this approach encourages the responsiveness of the reconstructed light spectrum such that it reacts quickly when a major change in intensity is detected across the wavelength while otherwise maintaining stability. This process preserves more edge information, such as square wave-like outputs. The hyperparameter $c_{4,1}$ promotes continuity and smoothness of the reconstructed light spectrum. This term is called QV in the literature[56]. In the present context, this approach facilitates Gaussian-wave-like outputs. The hyperparameters $c_{3,p}$ ($p \geq 2$) control variants of TV, while $c_{4,q}$ ($q \geq 2$) control variants of QV. They introduce larger step sizes in TV and QV and should be set according to the spacing of the input light spectra in training. In practice, only a few of those terms are needed, with the other terms set to 0.

Notably, our machine learning algorithm represents a generalization of various algorithms, such as least squares, LASSO, ridge regression, elastic net, fused LASSO, TV, and QV, and thus our model adapts to different practical cases by using different hyperparameters. When $c_1 = c_2 = c_{3,p} = c_{4,q} = 0$ ($p \geq 1$, $q \geq 1$), the optimization reduces to least squares. When $c_2 = c_{3,p} = c_{4,q} = 0$ ($p \geq 1$, $q \geq 1$), the optimization becomes LASSO. When $c_1 = c_{3,p} = c_{4,q} = 0$ ($p \geq 1$, $q \geq 1$), the optimization involves ridge regression. When $c_{3,p} = c_{4,q} = 0$ ($p \geq 1$, $q \geq 1$), the optimization appears as an elastic net, which is the combination of LASSO and ridge regression[53]. When $c_2 = c_{3,p} = c_{4,q} = 0$ ($p \geq 2$, $q \geq 1$), the optimization comes to be fused LASSO. When $c_1 = c_2 = c_{3,p} = c_{4,q} = 0$ ($p \geq 2$, $q \geq 1$), the optimization is TV. When $c_1 = c_2 = c_{3,p} = c_{4,q} = 0$ ($p \geq 1$, $q \geq 2$), the optimization is QV. Generally, the larger the hyperparameters ($c_1$, $c_2$, $c_{3,p}$, and $c_{4,q}$) are, the more robust the method is to noise. Nevertheless, because the use of too large hyperparameters may filter out weak and sharp signals, proper selection of hyperparameters is important for computational spectrometers to simultaneously contain noise and maintain spectral resolution. In the implementation, CVXPY[57,58], an open-source Python package for convex optimization problems, is used to determine the optimal $w_j$. With the calculated $w_j$, the input light spectrum $\widetilde{\mathbf{x}}(\lambda)$ can be reconstructed by using the training light spectra $\mathbf{x}_j(\lambda)$, as shown in Eq. (2). A diagram of our machine learning-based spectrum reconstruction procedure is shown in Figure S6.

In our experiments, a tunable optical monochromator with a broadband halogen tungsten light source (WDG15-Z, Beijing Optical Century Instrument Co., Ltd., China), a xenon lamp light source (BBZM-3, Anhui Bobei Lighting Electrical Appliance Factory, China), a monochrome camera (ASI178MM camera, Suzhou ZWO CO., LTD., China) with a 14-bit CMOS image sensor with 3096 × 2080 pixels (Sony IMX178 sensor, Sony Group Corporation, Japan), and a commercial optical spectrometer (USB4000, Ocean Optics, USA) were used to construct a setup for training and testing the plasmonic microfilter array-based computational spectrometer, as shown in Figure S7. For

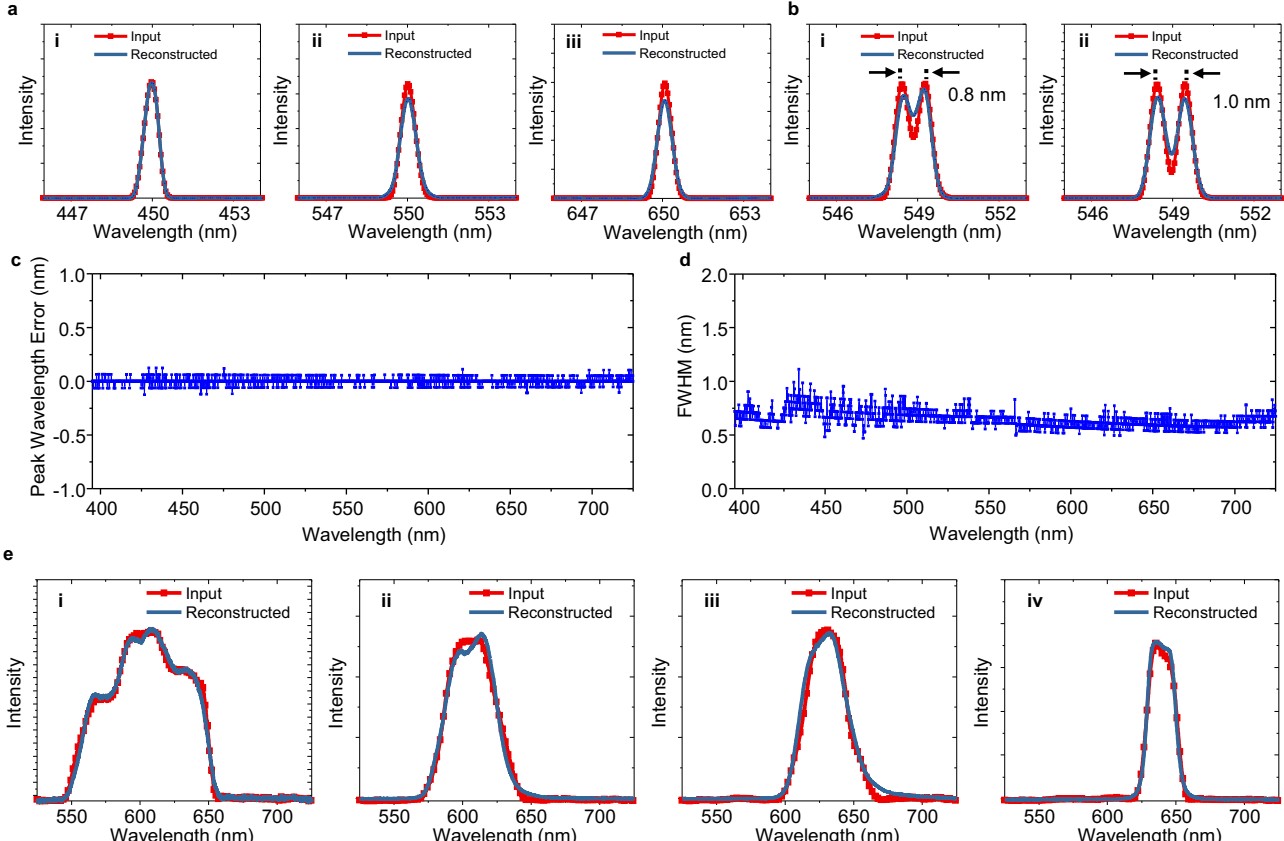

**Fig. 5 | Testing results of the miniature computational spectrometer. a** Testing results for single-peak spectrum inputs: (i–iii) the peak wavelengths of input single-peak spectra are ~450, ~550, and ~650 nm, respectively. **b** Testing results for dual-peak spectrum inputs: (i, ii) the wavelength separations of input dual-peak spectra are 0.8, and 1.0 nm, respectively. **c** The root-mean-square errors (RMSEs) of the

peak wavelengths of the reconstructed spectra for narrow single-peak spectrum inputs. **d** The FWHMs of the reconstructed spectra for narrow single-peak spectrum inputs. **e** Testing results for some broadband input spectra: (i–iv) the bandwidths of the broadband input spectra are ~85, ~40, ~37, and ~22 nm, respectively.

ease of installation, the fabricated microfilter array was not directly attached to the CMOS image sensor but was instead installed on the imaging position of a lens (DTCM110-16.6, VICO Technology Co., Ltd., China) that can clearly project the light pattern passing through the microfilter array onto the CMOS image sensor. The light output from the tuneable monochromator is divided into two light beams by a beam splitter. One light beam is monitored by a commercial spectrometer to obtain $x_j$, while the other light beam passes through the microfilter array, and the resulting light pattern generated by the microfilter array is recorded by the monochromatic camera to obtain $y_j$. These monochrome images that are measured by the camera, where the dark field and bias response were carefully corrected to enhance the signal-to-noise ratio (SNR), were pretreated with pixel binning and data filtering for noise reduction. Typical image data after denoising processes for spectral reconstruction are shown in Fig. S8.

Figure 5 shows the test results of the computational optical spectrometer, in which the control hyperparameters $c_1, c_2, c_{3,1}, c_{3,5}, c_{3,9}$, and $c_{4,1}$ in the objective function (Eq. (5)) are 600, 0.1, 0.1, 0.02, 0.05 and 0.06, respectively. The other hyperparameters, i.e., $c_{3,p}$ ($p = 2, 3, 4, 6, 7, 8$, or $p \geq 10$) and $c_{4,q}$ ($q \geq 2$), are all zero. In the first group of tests, a series of single-peak light spectra with a full width at half maximum (FWHM) of ~0.54 nm was used for training and testing the spectrometer. As shown in Fig. 5a, the computational spectrometer can measure peak wavelengths very well at wavelengths of 450, 550, and 650 nm. However, the spectral profile reconstructed at the short wavelength of 650 nm is slightly wider than the input light spectrum, and better results are achieved at wavelengths of 450 and 550 nm. The spectrum-resolving ability of the spectrometer was tested by using a

time-elongated exposure of two quickly generated adjacent single-peak light spectra. Figure 5b shows that the spectrometer can discriminate two adjacent spectral peaks with a wavelength separation of ~0.8 nm. Further testing of the spectrometer's performance over the whole visible range was conducted by sweeping such single-peak light spectra over wavelengths ranging from 395 nm to 725 nm with a step of 0.2 nm. Figure 5c shows the root-mean-square errors (RMSEs) of the peak wavelengths of the measured spectra. Figure 5c shows that the spectrometer can measure peak wavelengths accurately over the whole testing wavelength range (i.e., 395–725 nm), and its error is approximately 0.03 nm. The results of the measured FWHMs of the peak wavelengths of the spectra are shown in Fig. 5d. The average value of the FWHMs is 0.65 nm, which is close to the FWHM of the input spectrum, i.e., 0.54 nm.

The ability of the spectrometer to measure broadband light spectra with asymmetric profiles was also tested. Here, the light spectra were generated by using the broadband light source of the monochromator together with a few bandpass filters. For fast training and measurement, the spectrometer was trained with a series of broader single-peak spectra with an average FWHM of ~8.6 nm. Correspondingly, the best suitable control parameters for spectrum reconstruction were altered to $c_1 = 1.41 \times 10^5$, $c_2 = 1.4$, $c_{3,1} = 1.09 \times 10^7$, $c_{3,5} = 1.14 \times 10^5$, $c_{3,9} = 4.6$ and $c_{4,1} = 4000$. The other hyperparameters are all zero. The resulting spectral reconstructions are shown in Fig. 5e. The spectrometer can measure and reconstruct broadband spectra of different profiles well, and smooth or abrupt changes in the spectral curves were reconstructed. For the four input spectra under test, the cosine similarities of their reconstructed spectra are 0.9981, 0.9958,

0.9941, and 0.9983. Additional test results for the measurement of more broadband light spectra from OLEDs are given in Fig. S9. Another test of the measurement of narrowband and broadband input spectra using the same set of narrowband single-peak training spectra is shown in Fig. S10. These results show that spectrometers can generally measure various input spectra with different profiles.

## Discussion

Because of their large number of photosensors, CMOS/CCD camera-based computational spectrometers are widely regarded as among the most promising tools for achieving next-generation high-performance miniature/portable optical spectrometers. To achieve high resolution for such a computational spectrometer, the first step is to construct a thin-film transmissive spectrum-disparate microfilter array with high-density distinct spectral peaks or/and valleys. Here, we present a plasmonic nanoparticles-in-cavity microfilter array that provides a near-perfect solution to this challenge. Two independent parameters, i.e., the size of the AgNPs and the length of the FP cavity, can be harnessed to tune the transmission spectra of the microfilters. Compared to the scheme using pure FP cavity-based microfilters (i.e., without plasmonic nanoparticles)[35], this approach can efficiently increase the number of spectrum-disparate microfilters and thereby solve the challenge of ultrahigh-precision nanofabrication of a large number of different cavity lengths within a micrometer-scale thin film. Compared to the scheme using pure plasmonic nanoparticles (i.e., without FP microcavities), this approach can greatly enhance the spectral tailoring ability and avoid the tough requirement of the precise engineering of individual plasmonic nanostructures for the fabrication of a large-scale spectrum-disparate plasmonic microfilter array.

We experimentally demonstrated that both the size and length of the FP cavity of AgNPs can be precisely tailored by computer-controlled dynamic UV exposure technology. The light pixels of the UV light pattern are grouped to work parallelly in the patterning of microfilter elements, while the exposure dose, which is typically controlled according to the exposure time, is utilized to regulate nanoparticle size and tailor cavity length in the printing of AgNPs and the FP cavity, respectively. Due to implementing such a unique fabrication strategy, the proposed computational spectrometer has an important advantage, i.e., high scalability, over other previously reported methods (see the comparison in Table S1) and thus has great potential in the development of practical miniature computational spectrometers.

An evaluation of the critical role of the microfilter number in enhancing the spectrometer resolution is presented in Fig. S11. Here, two groups of single-narrow-peak light spectra with FWHMs of 0.61 and 0.44 nm were used to train and test a computational spectrometer based on a plasmonic microfilter array of 1440 elements, after which a portion of the CMOS outputs corresponding to a certain amount of microfilters were sequentially extracted for spectrum reconstruction. The achievable spectral resolution, represented by the achieved FWHM of the reconstructed spectral peaks, is dramatically enhanced with increasing microfilter number when the total number of filters is less than ~500. The increase in the number of microfilters gradually decreases when the total number of microfilters is greater than ~500, and this contribution becomes negligible until the total number of microfilters reaches ~1152 (for the case trained and tested with single-peak spectra with a FWHM of 0.44 nm) or ~1440 (for the case trained and tested with single-peak spectra with a FWHM of 0.61 nm).

The second most significant factor that may limit the performance of computational spectrometers, such as resolution, is the efficiency of spectrum reconstruction algorithms and the training dataset. Figures S12 and S13 present a comparison of the performances of different algorithms, including ridge, TV, LASSO, and our hybrid algorithm, on the reconstruction of spectra. Here, a series of single-peak input spectra with an average FWHM of 0.44 nm over the

wavelength range from 580 to 680 nm were used for training and testing. Compared with the results obtained by the LASSO, ridge, and TV algorithms, the ridge algorithm performed better in terms of peak wavelength error (Fig. S12(a−c)), while the LASSO algorithm had relatively better performance in terms of the FWHM (Fig. S13(a−c)). Inspired by these results, we developed a hybrid algorithm that is similar to LASSO (i.e., $c_1 = 600$, $c_2 = 0.1$, $c_{3,1} = 0.1$, $c_{3,5} = 0.02$, $c_{3,9} = 0.05$ and $c_{4,1} = 0.06$ in Eq. (5); the other hyperparameters are all zero.) for spectrum reconstruction. With these optimized parameters, the hybrid algorithm can achieve an average FWHM of 0.48 nm, which is highly consistent with the FWHM of the input spectrum. The RMSE between the input and reconstructed peak wavelengths is 0.018 nm, which also indicates the high accuracy of the spectrometer in terms of spectrum reconstruction.

A spectrometer trained with narrower input spectra also exhibited better performance in resolving two adjacent spectral peaks, as shown in Fig. S14. The spectrometer can resolve two spectral peaks well with a wavelength separation of 0.6 nm. This result is better than that achieved by the spectrometer trained with the input spectra with 0.54-nm wide input spectra, whose resolvable wavelength separation is ~0.8 nm, as shown in Fig. 5b(i). Using the spectral peak density information given in Fig. S3, the ultimate resolution of the computational spectrometer can be optimistically estimated to be 0.15–0.3 nm, which indicates that the performance of such a computational spectrometer may be further improved if a better reconstruction algorithm and training method are adopted.

Another remarkable factor that may limit the performance of computational spectrometers is noise, such as system errors and/or measurement noise, because the spectral reconstruction problem of a computational spectrometer is typically a seriously ill-conditioned system of equations whose solution is sensitive to perturbations coming from various types of noises[59,60]. To assess the susceptibility of the computational spectrometer to noise, we intentionally loosened the image data denoising process and then tested the effect of noise levels, in terms of the root mean square error (RMSE), on the achievable resolution. Figure S15 shows the corresponding test results for single-narrow-peak inputs with a spectral width of ~0.61 nm in the range of 527–725 nm. The achievable spectral resolution was enhanced from 2.16 to 0.71 nm when the RMSE was decreased from 3.13% to 0.31%. This noise-susceptible result is in line with those of previously reported computational spectrometers[21,22].

To demonstrate its potential applications, a smartphone-based computational spectrometer was constructed using a plasmonic microfilter array and assembled as shown in Fig. S16a. One can see that such a plasmonic microfilter array is very compatible with a smartphone because it makes full use of its image sensor and computing power to make portable computational spectrometers. The preliminary results obtained with the spectrometer are shown in Fig. S16b. One can see that the portable spectrometer can very well measure the broadband light spectrum with a bandwidth of ~50 nm. However, its performance in measuring narrow spectral peaks needs further improvement, which may be attributed to the relatively low contrast of the image sensor.

Therefore, to further improve the performance of computational spectrometers, especially in terms of spectral resolution, one needs to further improve the fabrication processes, e.g., increase the uniformity of AgNPs, to create better-quality plasmonic microfilter arrays with high-density spectral peaks or/and valleys uniformly distributed among elements; additionally, a better spectrum reconstruction algorithm that is less sensitive to noise should be pursued. The monochromator-based training setup can be further upgraded to provide narrower bandwidth light spectra for training high-resolution spectrometers. In addition, a better image sensor with high contrast and a highly linear response is also needed for pursuing the ultimate performance of computational spectrometers. To further pursue a

broader range of applications, we can choose differential material platforms to fabricate FP microcavities with more distinct cavity lengths and to print different plasmonic nanoparticles, e.g., gold nanoparticles (AuNPs), with Mie resonance wavelengths that are longer than that of AgNPs, to construct a plasmonic microfilter array for high-resolution broadband computational spectrometers.

An AI-empowered high-resolution computational spectrometer based on a CMOS image sensor has been presented for measuring the input spectra of visible light. A plasmonic nanoparticles-in-cavity microfilter with strong coupling between the localized surface plasmon resonance of AgNPs and an optical FP microcavity was devised to flexibly regulate the transmission spectra by tuning the size and length of the FP microcavity for the creation of a large-scale scalable optical disparate microfilter array. Moreover, digital UV lithography-based fabrication processes have been established to fabricate a designed plasmonic microfilter array in which both the AgNP size and FP cavity length are precisely tuned by a computer-controlled dynamic UV exposure technique. A plasmonic microfilter array with 1152 elements was fabricated for integration with a commercial CMOS image sensor to construct a miniature computational spectrometer. With a machine learning-based training process, an AI-empowered miniature spectrometer has been demonstrated to measure different input spectra at an average spectral resolution of 0.65 nm over the whole visible light range. Such a highly scalable plasmonic microfilter array may pave the way for the development of high-performance miniature computational spectrometers for many applications, such as portable skin health monitoring and spectral imaging devices.

## Methods

### Materials
Silver nitrate, ethylene glycol, and titanium (IV) butoxide (97%) were purchased from Sigma Aldrich, Inc., USA. Shenzhen Huashi Technology Co., Ltd., China, provided nitric acid solution (1 mol/L aqueous solution). Shanghai Guangyi Chemical Co., Ltd., China, supplied dipentaerythritol hexaacrylate (DPHA) acrylate resin (Easepi 7300). The photoinitiator ethyl (2,4,6-trimethylbenzoyl)phenylphosphinate (TPO-L) was purchased from CNBM (Chengdu) Optoelectronic Materials Co., Ltd., China. The FC 4432 fluorocarbon surfactant was purchased from 3 M Co., USA. Isopropyl alcohol (IPA) and acetone were obtained from Anaqua Chemicals Supply, Inc., Ltd., USA. The Donghai County Zhongzheng Quartz Products Factory, China, provided quartz sheets with a thickness of 1 mm. All the materials were used as received without further purification. Deionized (DI) water with a resistance of 18 MΩ·cm was used in all the experiments.

### Sputtering of silver and silica nanolayers
The quartz substrates were cleaned using DI water, IPA, and acetone in an ultrasonic cleaner and then subjected to high-power plasma treatment for 300 seconds using a plasma cleaner (PDC-002-HP, Harrick Plasma, USA). After cleaning and treatment, the substrates were coated with an Ag/SiO$_2$ thin film using a sputtering system (Denton Vacuum, USA) with Ag and SiO$_2$ sputtering targets. The sputtering process was carried out with the following parameters: DC 50 W/Ar 30 sccm for the Ag layer (with a deposition rate of ~10.5 nm/min) and RF 200 W/Ar 30 sccm for the SiO$_2$ layer (with a deposition rate of ~3.3 nm/min).

### Sol-gel spin coating of the titanium dioxide nanolayer
Titanium (IV) butoxide was diluted with IPA to afford a 2 wt% solution. Then, 1 mol/L nitric acid aqueous solution was added at a volume ratio of 1:100. The final solution was then spin-coated on a quartz substrate capped with silver/silica nanolayers. The process consisted of spin coating at 500 rpm for 6 s followed by centrifugation at 3000 rpm for 30 s. The substrates were then baked at 110 °C for 10 min to remove the solvent and convert titanate to TiO$_2$ through hydrolysis in air.

### Direct printing of size-controlled AgNPs by precision photoreduction
A homemade trough was made of glass slide and coverslip to contain a 2% solution of silver nitrate in ethylene glycol. The quartz substrate spin-coated with a titanium dioxide nanolayer was then placed inside the glass trough for the printing of size-controlled AgNPs. From top to bottom, the structure has multiple layers, including the coverslip, the silver salt solution, and the quartz substrate. UV light irradiated the TiO$_2$ nanolayer on the quartz substrate through the coverslip and silver salt solution.

An in-house digital UV lithography setup built with a 365-nm UV light-emitting diode and a digital micromirror device (DMD) with 1920 × 1080 pixels (DLi6500, Texas Instruments, USA) was used to print the size-controlled AgNPs. Grayscale image data were loaded to the DMD to generate dynamic light patterns. The optical resolution of the system is approximately 297 nm, and its highest illumination intensity is 1803 mW cm$^{-2}$. The typical exposure time was 1–60 seconds. After the exposure process, the quartz substrate printed with AgNPs was rinsed sequentially with DI water and IPA and finally dried with a nitrogen gun.

### Printing of polymer FP cavities by digital grayscale photopolymerization technology
Easepi 7300 DPHA acrylate resin was mixed with 2% TPO-L photoinitiator. After diluting IPA to a 30 wt% concentration, 0.2 wt% of the FC 4432 fluorocarbon surfactant was added. The substrate was ultrasonically cleaned and then spin-coated with the photopolymer solution using a two-step spin coating process, i.e., 500 rpm for 6 s and 2000 rpm for 30 s. After spin coating, the substrate was baked on a hot plate at 80 °C for 2 min to remove the solvent and ensure proper adhesion of the photopolymer. Greyscale UV exposure was performed by using the same in-house digital UV lithography setup used in the printing of size-controlled AgNPs. The typical exposure time was 1–10 seconds. After exposure, the sample was developed by using acetone and IPA and finally dried by using a nitrogen gun.

### Numerical simulation
The spectral responses of the plasmonic nanoparticles-in-cavity microfilters were numerically simulated by using the commercial software COMSOL Multiphysics. AgNPs were modeled as periodic arrays in a square lattice. A linearly polarized plane wave was used as excitation light to illuminate the nanoparticles-in-cavity structure at normal incidence from the side of the quartz substrate.

### Spectrometer characterization and testing
Optical color images of the plasmonic microfilters were taken by using a commercial metallurgical microscope (L3203, Guangzhou LISS Optical Instrument Co., Ltd., China) and a camera (A7R2, Sony Group Corporation, Japan). Field-emission scanning electron microscopy (MAIA3, TESCAN, Czech) was used to obtain microscopy images of the microfilter samples. The transmission and reflection optical spectra of the samples were measured by using UV–Vis optical fiber spectrometers (USB650 and USB4000, Ocean Optics, USA) with a 50x objective installed on a metallurgical microscope (L3203, Guangzhou LISS Optical Instrument Co., Ltd., China).

In addition to the custom-made plasmonic microfilter array, a C-mount bitelecentric lens (DTCM110-16.6; VICO Technology Co., Ltd., China) and a CMOS monocolour image camera (ASI178MM; integrated with a Sony IMX178 sensor and 14-bit apparent diffusion coefficient (ADC)) were used to construct the computational spectrometer. The spectrum-tunable light sources used in the training and testing of the computational spectrometer were a tungsten halogen lamp plus a grating-based monochromator (WDG15-Z, Beijing Optical Century Instrument Co., Ltd., China) and a xenon lamp light source (BBZM-3, Anhui Bobei Lighting Electrical Appliance Factory, China).

**Reporting summary**

Further information on research design is available in the Nature Portfolio Reporting Summary linked to this article.

## Data availability

Source data are provided in this paper.

## Code availability

The codes that support the findings of this study are available in [repository name e.g., "figshare"] with the identifier(s) [data DOI].

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

## Acknowledgements

This work was fully supported by a grant from the Research Grants Council of the Hong Kong Special Administrative Region (Grant No. 15208120). G.L. gratefully acknowledges the support of the National Science Foundation (DMS-2053746, DMS-2134209, ECCS-2328241, and OAC-2311848).

## Author contributions

A.P.Z. and G.L. directed the research and acquired the project funding. Y.Z. and A.P.Z. conceived the plasmonic microfilter array for computational spectrometers. Y.Z. conducted the numerical simulation and optimization. Y.Z. and H.W. fabricated the plasmonic microfilter array. Y.Z. and J.W. constructed the spectrometer training setup. S.Z. and G.L. designed the machine learning algorithms. S.Z. and Y.Z. developed the codes for spectrum reconstruction. Y.Z., S.Z., G.L. and A.P.Z. analyzed the data and wrote the original draft. All the authors contributed to reviewing and editing the paper.

## Competing interests

The authors declare no competing interests.
