## [Peer Review File · Nature Communications]

Miniature computational spectrometer with a plasmonic nanoparticles-in-cavity microfilter arrayEditorial Note: Parts of this Peer Review File have been redacted as indicated to remove third-party material where no permission to publish could be obtained.

REVIEWER COMMENTS

Reviewer #1 (Remarks to the Author):

Efforts to miniturize spectroscopic systems are currently a very timely area - but such miniturization usually leads to inaccurately reconstructed spectra. Attaining accurate reconstruction by simply exploiting close spectral resonances is a key current and ongoing objective of the broader field of optical spectroscopy.

What this paper reports is a novel artificial-intelligence based method for obtaining accurate such spectra - for miniturized devices. Although the training time for the AI method may be long, the time for spectrum-reconstruction is efficient once the overall model is calibrated.

The graded plasmonic-nanoparticles based microfilter array that the authors use is an interesting scheme, allowing for localizing different incident light-wavelengths to different spatial points along the structure, thereby being very broadband and potentially practical.

Overall, this is a well-written, high-quality work, combining nicely several timely areas (graded broadband plasmonics, optical spectroscopy and polarimetry, and AI-based systems) to attain a useful functionality (described above).

As such, it makes an interesting / novel contribution to the broader field, meriting potential publication in NCOMMs after, from my perspective, the authors address the following points:

1. Is the used AI method guaranteed to converge to a global (rather than 'only' local) error-minimum point? That would be interesting to know, in order to assess the generality and general-reliability of the method (here, it does appear to be ok).

2. What is the maximum spectral resolution that can be attained with this scheme (on the reported structure)? Also, what is here the (well-known) figure-of-merit for the reported spectro-polarimeter?

3. How much susceptible (or not) to noise is the scheme, and why?

4. It should be interesting for the journal's broad readership to compare this scheme with other recent similar schemes / papers (e.g., Nature Communications 14, Article number: 1902 (2023)) for spectral and polarimetric analyses (e.g., in terms of fabrication requirements, scalability, etc) in order to even more clearly outline and emphasize the present scheme's comparative merits (or potential limits). Also, related to this, since the present scheme exploits a scheme quite reminiscent of the 'rainbow grading effect', I think that Nature Communications 14, Article number: 1902 (2023), Nature 450, 397-401 (2007), etc, should be cited.

Reviewer #2 (Remarks to the Author):

The authors report plasmon-integrated filter-based spectrometers. It seems to me not very clear about the advantages of using plasmon inside a cavity.

Here are my comments:

1. The authors should discuss the experimental advantages of devices compared to the methods with pure cavity-based filters (i.e., without plasmonic nanoparticles) and pure plasmonic nanoparticles (i.e., without cavities).

2. The operation bandwidth is around 100nm, which is quite narrow. I suggest the authors fully demonstrate the operation to at least cover the visible range. And also suggest discussing how to push them for a broader range.

3. The authors should fully discuss the optical resolution limits. How to achieve a better resolution?

4. It is also suggested that the authors to carry out some application demonstrations if possible.
5. It is also suggested to make a table to compare the performance with the state-of-the-art demonstrations and all key demonstrations with filter-based methods.
6. It will also be nice to fairly discuss the disadvantages of the methods in the main text.
7. The discussion of the key functions of AI is missing. Also, it is suggested to upload all of their code to a public website (Github and Zenodo).

Reviewer #3 (Remarks to the Author):

This manuscript presents a study on a miniaturized computational spectrometer based on a plasmonic nanoparticles-in-cavity microfilter array and a machine-learning-based reconstruction algorithm. To achieve high-efficient spectral encoding, the diameter of the AgNPs as well as the thickness of the intermediate polymer altered. The aforementioned changing parameters can be precisely controlled by UV exposure doses. Moreover, the reconstructed spectra are well-matched with the references. However, after careful consideration, I believe the manuscript does not meet the criteria for publication in Nature Communications primarily due to a lack of novelty.

Here are my remarks:

- 1) The fabrication method for the spectral encoders is similar to the one described in the published article "Xuan, Z.; Wang, Z.; Liu, Q.; et, al. Short-Wave Infrared Chip-Spectrometer by Using Laser Direct-Writing Grayscale Lithography. *Adv. Opt. Mater.* 2022, 10(19), 2200284". Introduction of silver nanoparticles does not significantly enhance the complexity of the spectral features, i.e. number of peaks or valleys.
- 2) The authors extensively discuss the spectral peak position and peak value distribution of the plasmonic nanoparticles-in-cavity microfilter array. However, many researches have proved that the primary requirement for the reference spectra is low correlation. From my perspective, it would be beneficial if the authors concentrate on the complexity of the reference spectra and provide the numerical value of the correlation coefficient of the

adopted spectral encoders.

3) The authors employed a deep learning-based reconstruction network, vivid descriptions of the proposed network/algorithm should be provided using figures, to make it easier for the readers to understand.

4) The authors spend much of the sections (including Figure 3) to illustrate the fabrication of the sized-controlled AgNPs, which is described in detail by the authors' previous works (Zhang Y, Zhang Q, Ouyang X, Lei DY, Zhang AP, Tam H-Y. Ultrafast light-controlled growth of silver nanoparticles for direct plasmonic color printing. *ACS nano* 2018, 12(10): 9913-9921; Zhang Y, Liang Z, Zhang AP, Tam HY. Direct Printing of Micropatterned Plasmonic Substrates of Size-Controlled Gold Nanoparticles by Precision Photoreduction. *Adv Opt Mater* 2021, 9(1): 2001368).

5) 1152 spectral encoders are adopted in the proposed miniaturized spectrometer, the number of the encoders seem too much, and the information may contain a significant amount of redundancy for spectral reconstruction. The excessive number of spectral encoders is in contradiction with the proposed facile fabrication method.

6) The manuscript should provide more experimental data to support the excellence of the proposed miniaturized spectrometer, such as spectral imaging of objects.

Revisions according to the comments by Reviewer 1:

General comment: “Efforts to miniturize spectroscopic systems are currently a very timely area - but such miniturization usually leads to inaccurately reconstructed spectra. Attaining accurate reconstruction by simply exploiting close spectral resonances is a key current and ongoing objective of the broader field of optical spectroscopy.

What this paper reports is a novel artificial-intelligence based method for obtaining accurate such spectra - for miniturized devices. Although the training time for the AI method may be long, the time for spectrum-reconstruction is efficient once the overall model is calibrated.

The graded plasmonic-nanoparticles based microfilter array that the authors use is an interesting scheme, allowing for localizing different incident light-wavelengths to different spatial points along the structure, thereby being very broadband and potentially practical.

Overall, this is a well-written, high-quality work, combining nicely several timely areas (graded broadband plasmonics, optical spectroscopy and polarimetry, and AI-based systems) to attain a useful functionality (described above).”

Reply:

Thanks very much for the reviewer's positive general comment on our work. We fully agree with the reviewer's prospect on miniaturized spectrometers and believe that our proposed scheme is very promising for developing broadband miniature spectroscopic systems for practical uses.

Comment 1: “Is the used AI method guaranteed to converge to a global (rather than 'only' local) error-minimum point? That would be interesting to know, in order to assess the generality and general-reliability of the method (here, it does appear to be ok).”

Reply and revision:

Yes, our AI method using a divide-and-conquer algorithm can converge to a global minimum. It has very little assumption on the property of the objective function and splits a large optimization problem into a series of subproblems, each of which has a closed-form solution.

Specifically, a loss function Eq. 5 has been carefully designed, whose minimization problem is a convex function problem that has only one local minimum which is also a global minimum, in our AI method.

When $c_2 > 0$, the quadratic penalty term in the loss function Eq. 5 makes the whole function strongly convex, e.g., the matrix $\nabla^2 Loss - 2c_2 I$ is positive semidefinite for all elements except a subset of measure zero, and it therefore has a unique global minimum and no other local minimum. In experiments, we used the Python-embedded modeling language package CVXPY to solve this optimization problem. CVXPY called an operator splitting quadratic program (OSQP, v0.6.2) solver, which is most specialized to this type of problems. It runs the alternating direction method of multipliers (ADMM) algorithm for convex quadratic programs which can converge to the global minimum from an arbitrary initial point.

We have revised our manuscript to clarify these points as:

“From the optimization perspective, the quadratic penalty term in the loss function Eq. 5 makes the whole function strongly convex⁴⁷, e.g., the matrix $\nabla^2 Loss - 2c_2 I$ is positive semidefinite for all elements except a subset of measure zero, which equips the algorithm many merits such as (i) unique global minimum without other local minimum and (ii) fast convergence.”

(Line 303, Page 11 ~ Line 307, Page 11)

Comment 2: “What is the maximum spectral resolution that can be attained with this scheme (on the reported structure)? Also, what is here the (well-known) figure-of-merit for the reported spectropolarimeter?”

Reply and revision:

As spectral resolution is a key figure-of-merit (FoM) of spectrometers, we have used two methods to test the spectral resolution of the computational spectrometer. Using the first classical approach, i.e., testing the minimum wavelength difference between two narrow spectral lines that can be resolved, the best resolution can be attained is ~ 0.79 nm as shown in Fig. 5b, in which a series of single-peak light spectra with the full-width at half-maximum (FWHM) of ~ 0.65 nm are used for training and testing. Remarkably, the resolution of this computational spectrometer may vary slightly if different light spectra with different FWHM are used for training. As shown in Fig. S13b, the best resolution can be attained of the computational spectrometer changes to 0.6 nm when single-peak light spectra with the FWHM of ~ 0.45 nm are used for training and testing.

Using a second common method, i.e., testing the resolution via measuring the FWHM of input narrow spectral lines, the best resolution of our computational spectrometer is ~ 0.64 nm achieved in the wavelength range from 690 nm to 810 nm when the input spectra with the FWHM of ~ 0.59 nm were used in the training and testing. In the other two wavelength regions, i.e., from 445 nm to 560 nm, and from 560 nm to 690 nm, their average values of measured FWHMs are 1.01 nm and 0.74 nm, respectively. Notably, the measured resolution of the computational spectrometer varies also when the light spectra with narrower FWHM are used for training. As shown in Fig. S12d, the best resolution can be attained changes to 0.49 nm in the wavelength range from 580 nm to 680 nm when a series of single-peak light spectra with the FWHM of ~ 0.45 nm are used for training and testing.

Due to the limitation of testing instruments, we are not able to have narrower single-peak light spectra to test the best resolution of the computational spectrometer. If assume that a better training setup with e.g., narrower input spectra, better imaging device, further optimized reconstruction algorithm, and more

powerful computing power, the ultimate resolution of the spectral spectrometer may close to the density of spectral peaks in the range of operation. Fig. S3a shows the distribution of spectral peaks of our microfilter array in terms of peak number in each 15 nm interval. Using such spectral peak density information, the ultimate resolution of the computational spectrometer may approach to 0.15 ~ 0.3 nm.

Thanks to a near-random distribution of silver nanoparticles (AgNPs), this plasmonic microfilter array-based computational spectrometer exhibits very low polarization dependence. **Fig. R1** shows the measured dependence of the transmittance of a typical plasmonic filter on polarization angle and the measured polarization dependence loss of 160 microfilters. It can be seen that such the transmittance of such plasmonic microfilters have a very small difference for light inputs whose polarization angles vary from 0° to 180°. The average polarization dependence of 160 microfilters is about 0.086 dB. Such a very low polarization independence may enable the computational spectrometer to achieve a good result even in light polarization-varying environments.

Figure R1. Polarization-dependence measurement results of plasmonic microfilters. **a.** Normalized transmittance of a typical filter for 0° to 180° linearly polarized light. The inset is the transmission spectra of a typical filter for 0° and 90° linearly polarized light. **b.** Measured polarization-dependence loss of 160 filters. The inset is the statistical distribution of polarization-dependent loss.

We have added Fig. R1 into the supporting information as Figure S4 and revised the manuscript to include an optimistic estimation of the ultimate resolution of the computational spectrometer and a discussion of polarization dependence of plasmonic microfilters as:

“If use the spectral peak density information given in Figure S3, the ultimate resolution of the computational spectrometer may be optimistically estimated to 0.15 ~ 0.3 nm, nevertheless the actual achievable resolution is subject to many factors, such as stability of input light, noises and resolution of imaging sensor as well as training and reconstruction algorithms.”

(Line 441 ~ 445, Page 14)

“The polarization dependence of such plasmonic microfilters was measured with a polarization-switchable input light beam, as given in Figure S4. The transmittance of plasmonic microfilters has a very small change when the polarization of the linearly polarized input light varies from 0 to 180°. The measured polarization dependence is about 0.086 dB, which indicates that the computational spectrometer is not sensitive to the polarization of input light.”

(Line 251 ~ 255, Page 9)

Comment 3: “How much susceptible (or not) to noise is the scheme, and why?”

Reply and revision:

Yes, noise is one of the significant factors that limit the performance of computational spectrometers. To quantitatively assess the susceptibility of our computational spectrometer to noise, we deliberately relaxed the noise reduction processing for the light intensity values after passing microfilters, quantified by the root mean square error (RMSE), and then tested its effect on the spectral resolution through monitoring the full-width half maximum (FWHM) of reconstructed narrow spectral peaks that has a spectral width of ~0.6 nm in the range of 525 ~ 675 nm. As shown in Figure R2, the testing result revealed that the spectrometer’s spectral resolution in terms of FWHM is proportional to the noise level. When the RMSE is depressed from 3.13% to 0.31%, the reconstructed spectral resolution can be enhanced from 2.08 nm to 0.7 nm. The result is in line with previously reported computational spectrometers^{18,19}

Figure R2. Testing results about the effect of measurement noise on the spectrometer's resolution. **a**, Average FWHM as a function of RMSE. **b**, Reconstruction result with the RMSE of 3.13%. **c**, Reconstruction result with the RMSE of 0.31%. The spectrometer was trained with the single-peak light spectra with the FWHM of 0.6 nm in the 525 ~ 675 nm range.

This is because these spectral reconstruction problems are typically seriously ill-conditioned equations, whose solution is sensitive to perturbations coming from various kinds of measurement errors and/or system noises^{R1, R2}. To depress noise effects, we have not only carefully corrected the image sensor's dark field and bias response, but also applied filtering processes to the output data from the image sensor. Moreover, our AI method has also included regularization terms to denoise, see Eq. (5). The larger the hyperparameters (c_1 , c_2 , c_3 , and c_4) are, the more robust to noise the method is. Nevertheless, as the use of too large hyperparameters may potentially filter out weak and sharp signals, a proper selection of hyperparameters is important for the computational spectrometer so as to simultaneously contain noise effects and maintain spectral resolution.

[R1] Neumaier, A. Solving ill-conditioned and singular linear systems: A tutorial on regularization. *SIAM Rev.* **40**, 636-666 (1998).

[R2] Tիրer, T., Girytes, R. Back-projection based fidelity term for ill-posed linear inverse problems. *IEEE Trans. Image Process* **29**, 6164-6179 (2020).

We have added Fig. R2 into the supporting information as Figure S14 and revised the manuscript to include the discussion of the noise effects on the computational spectrometer as:

“The monochrome images measured by the camera, whose dark field and bias response were carefully corrected to enhance signal-to-noise ratio (SNR), were pre-treated with pixel binning and data filtering for noise reduction. Typical image data after denoising processes for spectral reconstruction can be seen in Figure S8.”

(Line 342 ~ 345, Page 11)

“Generally, the larger the hyperparameters (c_1 , c_2 , c_3 , and c_4) are, the more robust to noise the method is. Nevertheless, as the use of too large hyperparameters may potentially filter out weak and sharp signals, a proper selection of hyperparameters is important for the computational spectrometer to simultaneously contain noise effects and maintain spectral resolution.”

(Line 321 ~ 325, Page 11)

“Another remarkable factor that may limit the performance of computational spectrometers is noises, such as system errors and/or measurement noises, because the spectral reconstruction problems of computational spectrometers are typically a seriously ill-conditioned system of equations whose solution is sensitive to perturbations coming from various kinds of noises^{53,54}. To assess the susceptibility of the computational spectrometer to noise, we intentionally loosened the image data denoising process and then tested the effect of noise levels, in terms of root mean square error (RMSE), on the achievable resolution. Figure S14 shows the testing results for single-narrow-peak inputs with a spectral width of ~0.6 nm in the range of 525 ~ 675 nm. The achievable spectral resolution was enhanced from 2.08 nm to 0.7 nm, when the RMSE was depressed from 3.13% to 0.31%, respectively. Such a noise-susceptible result is in line with previously reported computational spectrometers^{18,19}.”

(Line 446 ~ 456, Page 14)

The two reference papers [R1] and [R2] have been added to the list of references as [53] and [54], respectively.

Comment 4: “It should be interesting for the journal's broad readership to compare this scheme with other recent similar schemes / papers (e.g., Nature Communications 14, Article number: 1902 (2023)) for spectral and polarimetric analyses (e.g., in terms of fabrication requirements, scalability, etc) in order to even more clearly outline and emphasize the present scheme's comparative merits (or potential limits). Also, related to this, since the present scheme exploits a scheme quite reminiscent of the 'rainbow grading effect', I think that Nature Communications 14, Article number: 1902 (2023), Nature 450, 397-401 (2007), etc, should be cited.”

Reply and revision:

Thanks for the reviewer's kind suggestion. We agree to include these two papers and compare their scheme and working mechanism with our work. Moreover, we have prepared a table, i.e., Table R1, to clearly outline the differences between our work and other previously reported schemes.

Table R1 Comparison with recently reported computational spectrometers.

Publication information	Working mechanism	Operation mode	Operation range	Spectral resolution
Compact spectrometer based on a disordered photonic chip (Nature Photonics 2013)	Plasmonic nanohole	Transmission mode	1500 ~ 1525 nm	0.75 nm
A colloidal quantum dot spectrometer (Nature 2015)	Perovskite quantum dot	Transmission mode	390 ~ 690 nm	3.2 nm
Single-nanowire spectrometers (Science 2019)	Single compositionally engineered nanowire	Transmission mode	500 ~ 630 nm	7 ~ 8.5 nm
Single-shot on-chip spectral sensors based on photonic crystal slabs (Nature Communications 2019)	Photonic crystal slabs	Transmission mode	550 ~ 750 nm	1 nm
Compact CMOS spectral sensor for the visible spectrum (Photonics Research 2019)	Photonic crystal (PC) slabs	Transmission mode	400 ~ 700 nm	1 nm
Broadband perovskite quantum dot spectrometer beyond human visual resolution (Light: science & applications 2020)	Perovskite quantum dot	Transmission mode	250 ~ 1000 nm	1.6 nm
Neural Network-Based On-Chip Spectroscopy Using a Scalable Plasmonic Encoder (ACS Nano 2021)	Plasmonic nanohole	Transmission mode	480 ~ 750 nm	<3.29 nm
A Pearl Spectrometer (Nano Lett. 2021)	Naturally pearl nanostructure	Transmission mode	450 ~ 700 nm	7.4 nm
Deeply learned broadband encoding stochastic hyperspectral imaging (Light: science & applications 2021)	Multilayer film	Transmission mode	400 ~ 700 nm	5.2 nm
A wavelength-scale black phosphorus spectrometer (Nature Photonics 2021)	Black phosphorus	Transmission mode	2000 ~ 9000 nm	420 nm
3D-printed miniature spectrometer for the visible range with a 100 × 100 μm ² footprint (Light: Advanced Manufacturing 2021)	3D-printed micro-optics	Transmission mode	490 ~ 690 nm	9.2 nm
A Single-Dot Perovskite Spectrometer (Advanced Materials 2021)	LiCl-doped perovskite film	Transmission mode	350 ~ 750 nm	5.3 nm
Mass production-enabled computational spectrometers based on multilayer thin	Multilayer film	Transmission mode	500 ~ 849 nm	1 nm

films (Scientific Reports 2022)				
Ultraspectral Imaging Based on Metasurfaces with Freeform Shaped Meta-Atoms (Laser & Photonics Reviews 2022)	Metasurfaces with freeform shaped meta-atoms	Transmission mode	460 ~ 740 nm	0.5 nm
Dynamic brain spectrum acquired by a real-time ultraspectral imaging chip with reconfigurable metasurfaces (Optica 2022)	Reconfigurable metasurfaces	Transmission mode	450 ~ 750 nm	0.8 nm
Short-Wave Infrared Chip-Spectrometer by Using Laser Direct-Writing Grayscale Lithography (Advanced Optical Materials 2022)	FP cavities fabricated by laser direct writing	Transmission mode	900 ~ 1700 nm	2nm
Imaging-based intelligent spectrometer on a plasmonic rainbow chip (Nature Communications 2023)	Plasmonic rainbow chip	Reflection mode	400 ~ 800 nm	2 nm
This work	Plasmonic nanoparticles-in-cavity microfilter array	Transmission mode	445 ~ 810 nm	0.79 nm (0.49 nm for 580 ~ 680 nm)

Accordingly, we have revised the manuscript as:

“Many kinds of microfilters using different techniques, such as quantum dots^{18,19}, photonic crystals²⁰⁻²³, plasmonic encoder²⁴, plasmonic rainbow chip²⁵, metamaterials^{6,11,26,27}, liquid crystal²⁸, random structure^{29,30}, interference filters^{12,13,31,32}, and 3D-printed micro-optics³³, have been demonstrated to make computational spectrometers. In particular, nanophotonic methods that harness optical responses of nanostructures via material and geometric nanoengineering have become one of the most promising approaches. For instance, plasmonic 2D chirped gratings, in which the rainbow trapping effect is utilized to split wavelengths³⁴, have been used to make a dual-functional computational spectrometer for simultaneous spectroscopic and polarimetric analysis. However, the optical transmittance of such a metallic 2D chirped gratings is low and thus the computational spectrometer can work in reflection mode only, which may limit its further miniaturization for portable applications. Metamaterial/metasurface structures have also been demonstrated to fabricate optical microfilter arrays for computational spectrometers^{6,11}, and very promising results with sub-nanometer spectral resolution were demonstrated^{26, 27}.”

(Line 64, Page 2 ~ Line 77, Page 3)

“Owing to such a unique fabrication strategy, the proposed computational spectrometer renders an important advantage, i.e., high scalability, over other previously reported schemes (see the comparison in Table S1) and thus has great potential in the development of practical miniature computational spectrometers.”

(Line 407 ~ 410, Page 13)

The two mentioned-above papers have been added to the list of references as [25] and [34], respectively:

25. Tua, D., Liu, R., Yang, W., Zhou, L., Song, H., Ying, L., Gan, Q. Imaging-based intelligent spectrometer on a plasmonic rainbow chip. *Nat. Commun.* 2023, **14**: 1902.

34. Tsakmakidis, K.L., Boardman, A.D., Hess, O. 'Trapped rainbow' storage of light in metamaterials. *Nature* 2007, 450: 397-401.

Table R1 has been added to Supporting Information as Table S1.

Revisions according to the comments by Reviewer 2:

Comment 1: “The authors should discuss the experimental advantages of devices compared to the methods with pure cavity-based filters (i.e., without plasmonic nanoparticles) and pure plasmonic nanoparticles (i.e., without cavities).”

Reply and revision:

Thank you for the reviewer's suggestion. According to the suggestion, we have revised the manuscript to include these discussions:

- i) Comparison to the method using pure cavity-based microfilters (i.e., without plasmonic nanoparticles)

“Compared to the scheme using pure FP cavity-based microfilters (i.e., without plasmonic nanoparticles)³², it can solve the challenge of ultrahigh precision nanofabrication of a large number of different cavity lengths within a micrometer-scale thin film;”

(Line 396 ~ 399, Page 13)

- ii) Comparison to the method using pure plasmonic nanoparticles (i.e., without microcavities)

“Compared to the scheme using pure plasmonic nanoparticles (i.e., without FP microcavities), it can greatly enhance the spectral tailoring ability and avoid the tough requirement on the precise engineering of individual plasmonic nanostructures for fabrication of a large-scale spectrum-disparate plasmonic microfilter array.”

(Line 399 ~ 402, Page 13)

Comment 2: “The operation bandwidth is around 100 nm, which is quite narrow. I suggest the authors fully demonstrate the operation to at least cover the visible range. And also suggest discussing how to push them for a broader range.”

Reply and revision:

Thanks for the reviewer’s comment. Our computational spectrometer is designed to cover the whole visible light range. Due to the limitation of testing components and instruments, however, the broadest spectrum we tested in the experiment is about 100 nm. To test its spectral measurement performance over a broad wavelength range, we have used a series of single-narrow-peak spectra and tested the peak-wavelength measurement error and the spectral resolution in terms of minimum measurable FWHM in the wavelength range of 445 ~ 810 nm, as shown in Fig. 5c.

To reveal the ability of the spectrometer to measure broader spectra, here, we used three kinds of organic light-emitting diodes (OLEDs) to generate the input light spectra and measured their spectra by using the computational spectrometer, as shown in Fig. R3. It can be seen that the spectrometer can measure very well the input spectra with a bandwidth of 150 ~ 200 nm. The cosine similarities of the three results are 0.9762, 0.9695 and 0.9624, respectively. These results further verified the ability of our spectrometer to measure broad light spectra.

Figure R3. Testing results on the measurement of the broad light spectra from three different kinds of OLEDs. **a**, Yellow OLED. **b**, Cyan OLED. **c**, White OLED (from AMOLED screen of Motorola Edge 20 Pro). Their cosine similarities are 0.9762, 0.9695, and 0.9624, respectively.

One of the most important advantages of this computational spectrometer is its high scalability, which is attributed to its distinct plasmonic nanostructures/FP microcavity hybrid structure. To pursue further broader range, one may properly choose material platforms to fabricate relatively thicker FP microcavities with more distinct cavity lengths. Then, different plasmonic nanoparticles or nanostructures, e.g., gold nanoparticles (AuNPs) whose LSPR wavelength is longer than that of AgNPs, can be selected to make such a hybrid thin-film microfilter array for very broadband computational spectrometers.

To address these points, we added Fig. R3 into the supporting information as Fig. S9 and revised the manuscript as:

“For the four input spectra under test, the cosine similarities of their reconstructed spectra are 0.9998, 0.9950, 0.9965, and 0.9943, respectively. More testing results on the measurement of more broadband light spectra from OLEDs are given in Figure S9. These results show that the spectrometer can generally measure various input spectra with different profiles.”

(Line 383 ~ 386, Page 13)

“To further pursue broader range, one may properly choose material platforms to fabricate FP microcavities with more distinct cavity lengths, and print different plasmonic nanoparticles, e.g., gold nanoparticles (AuNPs) whose Mie resonance wavelength is longer than that of AgNPs, to make a

Comment 3: “The authors should fully discuss the optical resolution limits. How to achieve a better resolution?”

Reply and revision:

Thanks for the reviewer’s suggestion. We agree that it is important to fully discuss the resolution limits of the computational spectrometer. For CMOS/CCD camera-based computational spectrometers, the first important thing is to get a thin-film transmissive optical disparate microfilter array with high-density distinct spectral peaks or/and valleys. Such a thin-film microfilter array provides a wavelength-splitting or dividing function that can convert a light beam with a specific spectrum into a distinct gray-scale pattern on the image sensor. However, traditional lithography-based micro/nano-fabrication processes are good at massive duplication of the same microfilters but are not able or extremely expensive to fabricate a large-scale spectral disparate microfilter array. Our plasmonic nanoparticles-in-cavity microfilter array provides a near-perfect solution to this challenge. Two independent parameters, i.e., the size of AgNPs and the length of the FP cavity, have been harnessed to tune the transmission spectra of microfilters. However, the size uniformity of current directly printed AgNPs is still not perfect, which may limit the dynamic range or reduce the density of the spectral peaks of microfilters. It may deteriorate the wavelength splitting or resolving ability of the whole microfilter array and thus limit the spectral resolution of the computational spectrometer.

The second significant limit comes from the efficiency of spectrum reconstruction algorithms and the training dataset. We have compared the performance of several different algorithms, including Ridge, TV, LASSO, and our hybrid algorithm, on the spectrum reconstruction, see Figures S6 and S7. It can be seen that the Ridge algorithm performs better in terms of peak wavelength error, see Figure S6(a-c), while the LASSO algorithm has a relatively better performance in terms of the FWHM, see Figure S7(a-c). After proper optimization of hyperparameters, our hybrid algorithm achieved the best performance in terms of FWHM and peak wavelength error. Moreover, the training dataset also has a significant influence on the spectrometer’s resolution. As shown in Figure S8, a spectrometer trained with narrower input spectra obviously exhibits a better performance in resolving two adjacent spectral peaks.

Another limit on the spectrometer's spectral resolution is noises, such as system errors and/or measurement noises. This is because the spectral reconstruction problems of computational spectrometers are typically seriously ill-conditioned system equations whose solution is sensitive to perturbations coming from various kinds of noises. To assess the susceptibility of the computational spectrometer to noise, we intentionally loosened the image data denoising process and then tested the effect of noise levels, in terms of root mean square error (RMSE), on the achievable resolution. Figure S14 shows the testing results for single-narrow-peak inputs with a spectral width of ~ 0.6 nm in the range of 525 ~ 675 nm. One can see that the achievable spectral resolution was enhanced from 2.08 nm to 0.7 nm, when the RMSE is depressed from 3.13% to 0.31%, respectively.

To further improve the performance of the computational spectrometer, especially spectral resolution, one need not only further improve the fabrication processes, e.g., increase the uniformity of AgNPs, to make better-quality plasmonic microfilter array with uniformly distributed high-density spectral peaks

or/and valleys, but also pursue a better spectrum reconstruction algorithm that is less sensitive to noises. The monochromator-based training setup should be further upgraded to provide narrower bandwidth light spectra for training of high-resolution spectrometers. Beyond these factors, a better image sensor with high contrast and highly linear response is also needed for pursuing the ultimate performance of computational spectrometers.

We have revised the manuscript to address these points as:

“To achieve a high resolution for such a computational spectrometer, the first important thing is to make a thin-film transmissive spectrum-disparate microfilter array with high-density distinct spectral peaks or/and valleys.”

(Line 391 ~ 393, Page 13)

“The second most important things that may limit the performance of computational spectrometers, such as resolution, are the efficiency of spectrum reconstruction algorithms and the training dataset.”

(Line 423 ~ 425, Page 14)

“Another remarkable factor that may limit the performance of computational spectrometers is noises, such as system errors and/or measurement noises, because the spectral reconstruction problems of computational spectrometers are typically a seriously ill-conditioned system of equations whose solution is sensitive to perturbations coming from various kinds of noises. To assess the susceptibility of the computational spectrometer to noise, we intentionally loosened the image data denoising process and then tested the effect of noise levels, in terms of root mean square error (RMSE), on the achievable resolution. Figure S14 shows the testing results for single-narrow-peak inputs with a spectral width of ~0.6 nm in the range of 525 ~ 675 nm. The achievable spectral resolution was enhanced from 2.08 nm to 0.7 nm, when the RMSE was depressed from 3.13% to 0.31%, respectively. Such a noise-susceptible result is in line with previously reported computational spectrometers^{18,19}.”

(Line 446 ~ 456, Page 14)

“Therefore, to further improve the performance of the computational spectrometer, especially spectral resolution, one need not only further improve the fabrication processes, e.g., increase the uniformity of AgNPs, to make better-quality plasmonic microfilter array with high-density spectral peaks or/and valleys uniformly distributed among elements, but also pursue a better spectrum reconstruction algorithm that is less sensitive to noises. The monochromator-based training setup should be further upgraded to provide narrower bandwidth light spectra for training of high-resolution spectrometers. Beyond these factors, a better image sensor with high contrast and highly linear response is also needed for pursuing the ultimate performance of computational spectrometers.”

(Line 465 ~ 472, Page 15)

Comment 4: “It is also suggested that the authors to carry out some application demonstrations if possible.”

Reply and revision:

Thank you for the reviewer’s kind suggestion. The most important application of our scheme is to combine with CMOS/CCD image sensors to develop miniature portable spectrometers. To demonstrate such great potential, we here integrated our plasmonic microfilter array with commercial smartphone cameras, as shown in Fig. R4a, to develop smartphone-based computational spectrometers. In Fig. R4a

(i), a plasmonic microfilter array is attached to a CMOS image sensor (Sony IMX258) to make a portable spectrometer, while Fig. R4a (ii) shows the demonstration of a smartphone-based computational spectrometer with a plasmonic microfilter array. One can see that such a plasmonic microfilter array can make full use of the image sensor and the computing power of smartphones to develop portable computational spectrometers. Preliminary testing results of such a miniature spectrometer are given in Fig. R4b. One can see that the portable spectrometer can very well measure a broadband light spectrum with a bandwidth of ~ 50 nm. Its performance in the measurement of narrow spectral peak need further improvement, which may be attributed to the relatively low contrast of the image sensor.

Figure R4. Demonstration and testing of CMOS image sensor-based portable spectrometers with plasmonic microfilter array. **a**, Photos of the CMOS image sensor-based spectrometer (i) and the smartphone-based portable spectrometer (ii). **b**, Testing results of the home-made portable spectrometer shown in a(i) on the measurement of a single-narrow-peak spectrum (i) and a broadband light input (ii).

Moreover, we have also demonstrated the use of the computational spectrometer to measure the output light spectra of OLEDs, as shown in Fig. R3. These results indicated that such a plasmonic microfilter array may vigorously boost the development of practical portable computational spectrometers for daily use on a massive scale.

We have added Fig. R3 and Fig. R4 in the supporting information and revised the manuscript to address these points:

“For the four input spectra under test, the cosine similarities of their reconstructed spectra are 0.9998, 0.9950, 0.9965, and 0.9943, respectively. More testing results on the measurement of broad light spectra from OLEDs are given in Figure S9. These results show that the spectrometer can generally measure

various input spectra with different profiles.”

(Line 383 ~ 386, Page 13)

“To demonstrate its potential applications, a smartphone-based computational spectrometer using a plasmonic microfilter array is assembled as shown in Fig. S15a. One can see that such a plasmonic microfilter array is very compatible with a smartphone to make full use of its image sensor and computing power to make portable computational spectrometers. Preliminary testing results of the spectrometer are given in Fig. S15b. One can see that the portable spectrometer can very well measure a broadband light spectrum with a bandwidth of ~ 50 nm. However, its performance on measuring narrow spectral peaks needs further improvement, which may be attributed to the relatively low contrast of the image sensor.”

(Line 457, Page 14 ~ Line 464, Page 15)

Comment 5: “It is also suggested to make a table to compare the performance with the state-of-the-art demonstrations and all key demonstrations with filter-based methods.”

Reply and revision:

Thanks for the reviewer’s kind suggestion. Accordingly, we have added a table, i.e., Table R1, to the supporting information, i.e., Table S1, for comparing the key performance of all recently reported filter-based computational spectrometers.

Table R1 Comparison with recently reported computational spectrometers.

Publication information	Working mechanism	Operation mode	Operation range	Spectral resolution
Compact spectrometer based on a disordered photonic chip (Nature Photonics 2013)	Plasmonic nanohole	Transmission mode	1500 ~ 1525 nm	0.75 nm
A colloidal quantum dot spectrometer (Nature 2015)	Perovskite quantum dot	Transmission mode	390 ~ 690 nm	3.2 nm
Single-nanowire spectrometers (Science 2019)	Single compositionally engineered nanowire	Transmission mode	500 ~ 630 nm	7 ~ 8.5 nm
Single-shot on-chip spectral sensors based on photonic crystal slabs (Nature Communications 2019)	Photonic crystal slabs	Transmission mode	550 ~ 750 nm	1 nm
Compact CMOS spectral sensor for the visible spectrum (Photonics Research 2019)	Photonic crystal (PC) slabs	Transmission mode	400 ~ 700 nm	1 nm
Broadband perovskite quantum dot spectrometer beyond human visual resolution (Light: science & applications 2020)	Perovskite quantum dot	Transmission mode	250 ~ 1000 nm	1.6 nm
Neural Network-Based On-Chip Spectroscopy Using a Scalable Plasmonic Encoder (ACS Nano 2021)	Plasmonic nanohole	Transmission mode	480 ~ 750 nm	<3.29 nm
A Pearl Spectrometer (Nano Lett. 2021)	Naturally pearl nanostructure	Transmission mode	450 ~ 700 nm	7.4 nm
Deeply learned broadband encoding stochastic hyperspectral imaging (Light: science & applications 2021)	Multilayer film	Transmission mode	400 ~ 700 nm	5.2 nm
A wavelength-scale black phosphorus spectrometer (Nature Photonics 2021)	Black phosphorus	Transmission mode	2000 ~ 9000 nm	420 nm

3D-printed miniature spectrometer for the visible range with a 100 × 100 μm ² footprint (Light: Advanced Manufacturing 2021)	3D-printed micro-optics	Transmission mode	490 ~ 690 nm	9.2 nm
A Single-Dot Perovskite Spectrometer (Advanced Materials 2021)	LiCl-doped perovskite film	Transmission mode	350 ~ 750 nm	5.3 nm
Mass production-enabled computational spectrometers based on multilayer thin films (Scientific Reports 2022)	Multilayer film	Transmission mode	500 ~ 849 nm	1 nm
Ultraspectral Imaging Based on Metasurfaces with Freeform Shaped Meta-Atoms (Laser & Photonics Reviews 2022)	Metasurfaces with freeform shaped meta-atoms	Transmission mode	460 ~ 740 nm	0.5 nm
Dynamic brain spectrum acquired by a real-time ultraspectral imaging chip with reconfigurable metasurfaces (Optica 2022)	Reconfigurable metasurfaces	Transmission mode	450 ~ 750 nm	0.8 nm
Short-Wave Infrared Chip-Spectrometer by Using Laser Direct-Writing Grayscale Lithography (Advanced Optical Materials 2022)	FP cavities fabricated by laser direct writing	Transmission mode	900 ~ 1700 nm	2 nm
Imaging-based intelligent spectrometer on a plasmonic rainbow chip (Nature Communications 2023)	Plasmonic rainbow chip	Reflection mode	400 ~ 800 nm	2 nm
This work	Plasmonic nanoparticles-in-cavity microfilter array	Transmission mode	445 ~ 810 nm	0.79 nm (0.49 nm for 580 ~ 680 nm)

Comment 6: “It will also be nice to fairly discuss the disadvantages of the methods in the main text.”

Reply and revision:

Thanks for the reviewer’s suggestion. Yes, as the same with any other technology, our scheme also has some shortages: 1) The uniformity of directly printed size-controlled AgNPs is still not very high. The printing processes need further improvement e.g., nanoengineering of TiO₂ layer, to enhance the uniformity of AgNPs and thereby improve the sharpness of the spectral peaks in the transmission spectra of plasmonic microfilter array. 2) The current spectrum reconstruction algorithm is still quite sensitive to noises. Hyperparameters need to be carefully tuned for the reconstruction of different input spectra with different noise levels. Therefore, a better reconstruction algorithm that is less sensitive to noise needs to be further developed to improve user experience in practical applications.

We have revised the manuscript to include a short discussion of these shortages and indicate how to further improve the computational spectrometer:

“Therefore, to further improve the performance of the computational spectrometer, especially spectral resolution, one need not only further improve the fabrication processes, e.g., increase the uniformity of AgNPs, to make better-quality plasmonic microfilter array with uniformly distributed high-density spectral peaks or/and valleys, but also pursue a better spectrum reconstruction algorithm that is less sensitive to noises. The monochromator-based training setup should be further upgraded to provide narrower bandwidth light spectra for training of high-resolution spectrometers. Beyond these factors, a better image sensor with high contrast and highly linear response is also needed for pursuing the ultimate

Comment 7: “The discussion of the key functions of AI is missing. Also, it is suggested to upload all of their code to a public website (Github and Zenodo).”

Reply and revision:

Thanks for the reviewer’s suggestion. Our machine-learning-based AI method is established to enable a large-scale plasmonic microfilter array-based spectrometer to reconstruct the spectrum of an input light beam from CMOS image output. Considering that our plasmonic microfilter array has over one thousand elements and the image sensor has millions of pixels, we will have a large amount of data when we use a tunable monochromator-based setup to train the spectrometer via one-by-one scanning of a series of narrow spectral peaks over a broad wavelength range. While more training data is favored for achieving a higher resolution, it poses higher complexity and stability requirements on the spectrum reconstruction algorithm. This is especially a concern when large-scale arrayed microfilters with weak spectral orthogonality are used for wavelength splitting or dividing of input light beam. With such a microfilter array, the training process produces many similar image outputs. Hence, an elaborate AI method is needed to distinguish the discrepancy among a large number of similar image outputs and compare them with a measured CMOS reading for spectrum reconstruction.

More specifically, our AI method is designed to minimize the loss function Eq. 5 using a training dataset and thereby find optimal w_j for spectrum reconstruction. Its key functions include:

- 1) Least-squares term: minimizing the discrepancy between the calculated CMOS response of the reconstructed spectrum and the measured CMOS readings.
- 2) LASSO term: promoting sparser reconstruction (for example, low intensities are filtered out as noise, resulting in a narrow-peak light spectrum with zero intensity at most wavelengths).
- 3) L_2 term: reducing noise and adding stability to the algorithm when the training data are highly correlated. As large-scale arrayed microfilters with weak spectral orthogonality are used in our computational spectrometer, the training process produces many similar image outputs. Numerically, this causes the least-square regression problem in Eq. 4 ill-posed. Even when $n \approx m$, the similarity of CMOS outputs can cause non-identifiability and overfitting. The solution to this concern is to use L_2 norm regularization by setting $c_2 > 0$. This penalizes large components of the parameter vector w and hence smoothens the solution. From the optimization perspective, the quadratic penalty term in the loss function Eq. 5 makes the whole function strongly convex (e.g., the matrix $\nabla^2 Loss - 2c_2 I$ is positive semidefinite for all elements except a subset of measure zero), and it therefore has many properties, such as (i) unique global minimum and no other local minimum (ii) fast convergence.
- 4) Total variation term: enhancing the local constancy of the coefficient profile. In our context, it encourages the responsiveness of the reconstructed light spectrum, such that it reacts quickly when a major change of intensity is detected across the wavelength while trying to stay stable otherwise. It preserves more edge information such as square-wave-like outputs.
- 5) Quadratic variation term: promoting continuity and smoothness of the reconstructed light spectrum. In our context, it encourages Gaussian-wave-like outputs.

In short, the first key function is to ensure accurate reconstruction. The last four key functions are

regularization functions to denoise and ensure the robustness of the AI algorithm.

We have revised the manuscript to add more discussions about the method as:

“A machine-learning-based AI method was established to enable the spectrometer using a large-scale plasmonic microfilter array to reconstruct the spectrum of an input light beam from CMOS image output.”

(Line 262 ~ Line 264, Page 9)

“Therefore, the problem of finding $w_j, j=1, \dots, n$, for the spectrum reconstruction is transformed into solving the following optimization problem:

Minimize

$$Loss(w_1, \dots, w_n) = \sum_{k=1}^m \left(\tilde{y}_k - \sum_{j=1}^n w_j y_{j,k} \right)^2 + c_1 \sum_{j=1}^n |w_j| + c_2 \sum_{j=1}^n w_j^2 + c_3 \sum_{j=2}^n |w_j - w_{j-1}| + c_4 \sum_{j=2}^n (w_j - w_{j-1})^2, \quad (5)$$

where c_1, c_2, c_3 , and c_4 are non-negative hyperparameters. The first term is a least-squares term to minimize the discrepancy between the calculated CMOS response of the reconstructed spectrum and the measured CMOS readings. The last four terms are regularization terms to denoise and ensure the robustness of the machine learning algorithm.

The hyperparameter c_1 controls the L_1 norm of the parameter vector w and induces its sparsity such that w has fewer nonzero components. This term is also known as least absolute shrinkage and selection operator (LASSO)⁴⁵. In our context of spectrum reconstruction, larger c_1 promotes sparser reconstruction (for example, low intensities are filtered out as noise, resulting in a narrow-peak light spectrum with zero intensity at most wavelengths). The hyperparameter c_2 is for L_2 norm regularization of the parameter vector w . This term is also seen in ridge regression⁴⁶. The L_2 norm regularization reduces noise and adds stability to the algorithm when the training data are highly correlated. As large-scale arrayed microfilters with weak spectral orthogonality are used in our computational spectrometer, the training process produces many similar image outputs. Numerically, this causes the least-square regression problem in Eq. 4 ill-posed. Even when $n \approx m$, the similarity of CMOS outputs can cause non-identifiability and overfitting. The solution to this concern is to use L_2 norm regularization by setting $c_2 > 0$. This penalizes large components of the parameter vector w and hence smoothens the solution. From the optimization perspective, the quadratic penalty term in the loss function Eq. 5 makes the whole function strongly convex⁴⁷, e.g., the matrix $\nabla^2 Loss - 2c_2 I$ is positive semidefinite for all elements except a subset of measure zero, and it therefore has many good properties, such as (i) unique global minimum and no other local minimum and (ii) fast convergence.

The hyperparameter c_3 leads the term called TV, which often appears in fused LASSO^{19,48,49} to enhance local constancy of the coefficient profile. In our context, it encourages the responsiveness of the reconstructed light spectrum, such that it reacts quickly when a major change of intensity is detected across the wavelength while trying to stay stable otherwise. It preserves more edge information such as square-wave-like outputs. The hyperparameter c_4 promotes continuity and smoothness of the reconstructed light spectrum. The term is called quadratic variation in the literature⁵⁰. In our context, it encourages Gaussian-wave-like outputs.

Notably, our machine learning algorithm is a generalization of various algorithms, such as least

squares, LASSO, ridge regression, fused LASSO, TV, and QV, and thus adapts to different practical cases by using different hyperparameters. When $c_1=c_2=c_3=c_4=0$, the optimization reduces to least squares; when $c_2=c_3=c_4=0$, the optimization becomes LASSO; when $c_1=c_3=c_4=0$, the optimization is ridge regression; when $c_3=c_4=0$, the optimization appears as elastic-net, which is the combination of LASSO and ridge regression⁴⁷; when $c_2=c_4=0$, the optimization comes to be fused LASSO; when $c_1=c_2=c_4=0$, the optimization comes to be total variation. Generally, the larger the hyperparameters (c_1, c_2, c_3 , and c_4) are, the more robust to noise the method is. Nevertheless, as the use of too large hyperparameters may potentially filter out weak and sharp signals, a proper selection of hyperparameters is important for the computational spectrometer to simultaneously contain noise effects and maintain spectral resolution. CVXPY^{51,52}, an open-source Python package for convex optimization problems, is used to calculate w_j in the optimization. Since our loss function in Eqn. (5) is formulated as $2c_2$ -strongly convex, CVXPY converges to the unique global minimum when $c_2 > 0$. With the calculated w_j , the input light spectrum $\tilde{x}(\lambda)$ can be reconstructed by using the training light spectra $x_j(\lambda)$ as shown in Eq. (2). A diagram of our machine-learning-based spectrum reconstruction procedure is given in Figure S6.

(Line 283, Page 10 ~ Line 329, Page 11)

We will upload our codes to Figshare and Github once the paper is accepted for publication.

Revisions according to the comments by Reviewer 3:

General comment: “This manuscript presents a study on a miniaturized computational spectrometer based on a plasmonic nanoparticles-in-cavity microfilter array and a machine-learning-based reconstruction algorithm. To achieve high-efficient spectral encoding, the diameter of the AgNPs as well as the thickness of the intermediate polymer altered. The aforementioned changing parameters can be precisely controlled by UV exposure doses. Moreover, the reconstructed spectra are well-matched with the references. However, after careful consideration, I believe the manuscript does not meet the criteria for publication in Nature Communications primarily due to a lack of novelty.”

Reply:

We appreciate the reviewer’s critical comment on our work. However, we believe that our scheme using a directly printed plasmonic nanoparticles-in-cavity microfilter array to make an AI-empowered miniature spectrometer provides a totally new way for developing broadband practical portable spectrometers for extensive application. Compared with other proposed schemes, our scheme is a new-perfect solution to solve the challenge of making a thin-film large-scale transmissive optical disperse microfilter array with high-density distinct spectral peaks for CMOS/CCD-based miniature computational spectrometers. Moreover, we have accepted many valuable suggestions from all reviewers, which has further greatly improved the quality of our manuscript.

Comment 1: “The fabrication method for the spectral encoders is similar to the one described in the published article “Xuan, Z.; Wang, Z.; Liu, Q.; et, al. Short-Wave Infrared Chip-Spectrometer by Using Laser Direct-Writing Grayscale Lithography. Adv. Opt. Mater. 2022, 10(19), 2200284”. Introduction of silver nanoparticles does not significantly enhance the complexity of the spectral features, i.e. number of peaks or valleys.”

Reply and revision:

Thanks for the reviewer's comment. The recent paper mentioned by the reviewer proposed using pure cavity-length varying FP microcavities (without plasmonic nanoparticles) to make a microfilter array for miniature spectrometers. The main drawback of this scheme is its tough requirement on the ultrahigh precision nanofabrication of a large number of different cavity lengths within a micrometer-scale thin film. Therefore, using this scheme, one can achieve either a limited number of distinct microfilters or a very thick thin-film microfilter array that may hinder many practical applications. This should be the main reason why the paper demonstrated two spectrometers with only 50 microfilters and 20 microfilters and their spectral resolutions are 2 nm and 5 nm, respectively.

Our scheme using a plasmonic nanoparticles-in-cavity microfilter array aims to solve this challenge. A combination of n FP microcavity lengths and m AgNP sizes can provide $n \times m$ different microfilters and thus endows our thin-film microfilter device with high scalability for high-resolution broadband computational spectrometers.

Notably, the introduction of AgNPs doesn't have to increase the number of peaks or valleys for one single microfilter. It provides an effective means to further split spectral peaks or tune their peak wavelengths to make a large-scale optical disparate microfilter array. Indeed, we look for a large-scale microfilter array with high-density spectral peaks or/and valleys uniformly distributed among elements. In an ideal case, each microfilter could have one single distinct spectral peak or valley.

We have revised our manuscript to address these points as:

“Compared to the scheme using pure FP cavity-based microfilters (i.e., without plasmonic nanoparticles)³², it can solve the challenge of ultrahigh precision nanofabrication of a large number of different cavity lengths within a micrometer-scale thin film.”

(Line 396 ~ 399, Page 13)

Moreover, we have added the above-mentioned paper to the list of references and included their work in Table S1 for comparison.

32. Xuan, Z., Wang, Z., Liu, Q., Huang, S., Yang, B., Yang, L., Yin, Z., Xie, M., Li, C., Yu J., Wang, S., and Lu, W., Short-Wave Infrared Chip-Spectrometer by Using Laser Direct-Writing Grayscale Lithography. *Adv Opt Mater* 2022, **10**: 2200284.

Comment 2: “The authors extensively discuss the spectral peak position and peak value distribution of the plasmonic nanoparticles-in-cavity microfilter array. However, many researches have proved that the primary requirement for the reference spectra is low correlation. From my perspective, it would be beneficial if the authors concentrate on the complexity of the reference spectra and provide the numerical value of the correlation coefficient of the adopted spectral encoders.”

Reply and revision:

Thanks for the reviewer's invaluable suggestion. We agree that the primary requirement for the reference spectra of a spectral encoder for conventional spectrometers is low correlation. Ideally, all reference spectra are a series of sequentially arrayed non-overlapping spectral peaks whose cross-correlation coefficients are zero. However, the use of AI method for spectrum reconstruction can alleviate such a requirement and thus provide more flexibility in the design of spectral encoders, i.e., transmissive optical microfilter arrays in our case.

Yes, it is a very good idea to plot the cross-correlation coefficients of the transmission spectra of the fabricated plasmonic microfilter array for computational spectrometers. Figure R5a shows the cross-correlation coefficients of the transmission spectra of a fabricated plasmonic microfilter array with 1152 elements. A statistical distribution of these cross-correlation coefficients is given in Figure R5b. One can see that the majority (~ 70.6%) of these cross-correlation coefficients are between 0.2 and 0.6. This result signifies the necessity of the use of AI method for spectrum reconstruction in our computational spectrometer.

Figure R5. (a) Cross-correlation coefficients of the transmission spectra of a fabricated plasmonic microfilter array. (b) Statistical distribution of the cross-correlation coefficients.

We have added Fig. R5 into the supporting information as Figure S5 and revised the manuscript accordingly:

“The cross-correlation coefficients and their statistical distribution of the transmission spectra of a fabricated plasmonic microfilter are presented in Figure S5. One can see that the majority (~ 70.6%) of the cross-correlation coefficients are between 0.2 and 0.6. It signifies the necessity of the use of AI method for spectrum reconstruction in our computational spectrometer.”

(Line 255 ~ 259, Page 9)

Comment 3: “The authors employed a deep learning-based reconstruction network, vivid descriptions of the proposed network/algorithm should be provided using figures, to make it easier for the readers to understand.”

Reply and revision:

Thanks for the reviewer’s kind suggestion. Accordingly, we have prepared a diagram to show our machine-learning-based AI method for spectrum reconstruction, as shown in Fig. R6. The core of the AI method includes operator splitting quadratic program (OSQP) and alternating direction method of multipliers (ADMM), which is robust and fast with global convergence and can be applied to large-scale data.

Figure R6. Diagram of the machine-learning-based AI method for spectrum reconstruction.

We have added Figure R6 into the supporting information as Figure S6.

Comment 4: “The authors spend much of the sections (including Figure 3) to illustrate the fabrication of the sized-controlled AgNPs, which is described in detail by the authors’ previous works (Zhang Y, Zhang Q, Ouyang X, Lei DY, Zhang AP, Tam H-Y. Ultrafast light-controlled growth of silver nanoparticles for direct plasmonic color printing. ACS nano 2018, 12(10): 9913-9921; Zhang Y, Liang Z, Zhang AP, Tam HY. Direct Printing of Micropatterned Plasmonic Substrates of Size-Controlled Gold Nanoparticles by Precision Photoreduction. Adv Opt Mater 2021, 9(1): 2001368).”

Reply and revision:

Yes, our group published these two papers about the precision photoreduction technology for direct printing of the micropatterns of AgNPs and AuNPs before. To fabricate plasmonic nanoparticles-in-cavity microfilters, however, we noted that previous printing processes have to be further modified with the change from a transparent SiO₂ substrate to an opaque substrate with a silver mirror layer (for FP microcavity). Different from previous experiments in which UV light illuminated TiO₂ photocatalytic layer from the bottom SiO₂ substrate side, we here must let UV light illuminate through the silver salt solution. It causes two challenges:

- 1) UV light may directly trigger unwanted photoreduction reactions in the silver salt solution.
- 2) The scattering of UV light may become very strong with the growth of AgNPs, and then the UV pattern may become too blurry to print a well-defined micropattern of AgNPs.

To solve these challenges, we modified the previous material formulation. Ethylene glycol is used to replace water and glucose to act as both solvent and reducing agent. Previously used additive PVP is removed from the solution. Experiments revealed that such an adjustment can make the solution more resistant to intense UV light and enhance the stability of the printing process.

We have further revised the manuscript to clarify these points:

“Recently, we developed a precision photoreduction technology to additively print sized-controlled AgNPs or gold nanoparticles (AuNPs) using a digital UV exposure technique for plasmonic applications^{40,41}. With the use of an opaque substrate with a silver mirror layer (for FP microcavity), however, previous print processes can’t be directly applied, as UV light must illuminate TiO₂ photocatalytic layer through the silver salt solution instead of the bottom SiO₂ substrate side. It causes two challenges: i) UV light may directly trigger unwanted photoreduction reactions in silver salt solution; ii) The scattering of UV light may become strong with the growth of AgNPs, and then the UV pattern may become too blurry to print a well-defined micropattern of AgNPs. To overcome these challenges, we modified the previous material formulation. Ethylene glycol is employed to replace water and glucose to act as both solvent and reducing agent. Previously used additive PVP is removed from the solution. Experiments revealed that such an adjustment can lower a bit the activity of the silver salt solution and thus enhance the stability of the printing process.”

(Line 183 ~ 195, Page 6)

Comment 5: “1152 spectral encoders are adopted in the proposed miniaturized spectrometer, the number of the encoders seem too much, and the information may contain a significant amount of redundancy for spectral reconstruction. The excessive number of spectral encoders is in contradiction with the proposed facile fabrication method.”

Reply and revision:

Thanks for the reviewer’s thoughtful comment. To develop high-resolution broad-bandwidth miniature spectrometers, we fabricated a large-scale microfilter array with 1152 elements in this work. Such a large amount of microfilters can fully split or divide different wavelengths of a broadband input light beam and work together with a CMOS/CCD image sensor with millions of pixels to create a high-resolution gray-scale image for spectrum reconstruction. It is a record-breaking result of, but not in contradiction with our facile fabrication method.

On the other hand, we agree that it may create some quantity of redundancy for spectral reconstruction, and a quantitative assessment of the effect of microfilter number on the achieved spectrometer’s performance is worthy of study. Therefore, we used a miniature spectrometer with 1440 microfilters and conducted a supplemental test: two single-narrow-peak light spectra with the FWHM of 0.6 nm and 0.45 nm were used to train and test the computational spectrometer, and then a portion of CMOS outputs corresponding to a certain amount of microfilters are sequentially extracted for spectrum reconstruction. Figure R7 presents the testing result in which the FWHM of reconstructed spectra is used to indicate the achieved spectral resolution. One can see that there is a clear marginal effect of the dependence of the achievable spectral resolution of the spectrometer on microfilter number, i.e., the achievable spectral resolution can be dramatically enhanced with the increase of microfilter number when the total number of microfilters is below ~500; such a microfilter-number contributed enhancement will gradually decrease when the total number of microfilter is more than ~500, and its contribution becomes barely noticeable when the total number of microfilter is more than ~1152 (for the case trained and tested with the single-peak spectra with the FWHM of 0.45 nm) or ~1440 (for the case trained and tested with the single-peak spectra with the FWHM of 0.60 nm).

Figure R7. Testing results about the effect of microfilter number on the spectrometer's resolution. Two groups of single-peak spectra with the FWHM of 0.6 nm (red) and 0.45 nm (blue) were used in the training and testing of the spectrometer.

We have added Fig. R7 into supporting information as Figure S10 and revised the manuscript to discuss these points as:

“An evaluation of the critical role of microfilter number in enhancing spectrometer’s resolution is presented in Figure S10. Here two groups of single-narrow-peak light spectra with the FWHM of 0.6 nm and 0.45 nm were used to train and test a computational spectrometer based on a plasmonic microfilter array of 1440 elements, and then a portion of CMOS outputs corresponding to a certain amount of microfilters are sequentially extracted for spectrum reconstruction. One can see that the achievable spectral resolution, represented by the achieved FWHM of reconstructed spectral peaks, is dramatically enhanced with the increase of microfilter number when the total number is below ~500. The microfilter-number contributed enhancement will gradually decrease when the total number is more than ~500, and its contribution becomes barely noticeable until the total number of microfilters reaches ~1152 (for the case trained and tested with the single-peak spectra with the FWHM of 0.45 nm) or ~1440 (for the case trained and tested with the single-peak spectra with the FWHM of 0.60 nm).”

(Line 411, Page 13 ~ Line 422, Page 14)

Comment 6: “The manuscript should provide more experimental data to support the excellence of the proposed miniaturized spectrometer, such as spectral imaging of objects.”

Reply and revision:

Thanks very much for the reviewer’s prospective suggestion. We agree that spectral imaging is one of the very promising directions that our proposed miniaturized spectrometers could develop. However, in this work, we put our focus on the development of a plasmonic microfilter array and AI-based spectrum reconstruction method for miniaturized optical spectrometers. Nevertheless, to show the excellence of the proposed scheme, we have added more experimental data and demonstrated the use of a plasmonic microfilter array to develop smartphone-based portable spectrometers for e.g., skin health monitoring.

We have revised our manuscript to address these points as:

“To demonstrate its potential applications, a smartphone-based computational spectrometer using a plasmonic microfilter array is assembled as shown in Figure S15a.”

(Line 457 ~ 458, Page 14)

“Such a highly scalable plasmonic microfilter array may pave a new pathway to develop high-performance miniature computational spectrometers for many applications such as portable skin health monitoring and spectral imaging devices.”

(Line 491 ~ 493, Page 15)

Finally, we would like to thank the reviewers once again for their valuable comments, which have helped us to further improve the quality of the manuscript.

REVIEWER COMMENTS

Reviewer #1 (Remarks to the Author):

I have read the revised version of the manuscript, along with the revised SI and authors' replies.

The authors have done a good job in thoroughly revising their work, including a comparative table with other similar techniques, and an additional experimental demonstration of a CMOS-image-sensor based portable spectrometer with a plasmonic microfilter array, demonstrating the usefulness of their scheme.

From my perspective, the authors have satisfactorily addressed my previous comments to them, and have accordingly 'reflected' those crucial amendments in the revised main manuscript and SI.

As such, I feel that this updated, amended and extended version of the present timely work now makes a sufficiently strong 'case' as to merit publication in NCOMMs - I recommend acceptance as is.

Reviewer #2 (Remarks to the Author):

(1) It appears that the authors have not sufficiently provided experimental data to demonstrate the advantages or novelty of integrating plasmonic nanoparticles inside the cavities. It is recommended that they compare the Q factor and transmittance spectra of an FP cavity with and without plasmonic nanoparticles. This comparison will be crucial in showcasing the potential benefits of integrating plasmonic nanoparticles, such as improved Q factor and performance, which would justify their inclusion.

(2) The high resolution achieved by the authors may not be practically relevant, considering they used 2436 devices (filters) to achieve this level of resolution. Such a large number of devices would not be feasible for applications relying on CMOS sensor technology.

(3) The comparison table should incorporate more up-to-date results from state-of-the-art research to provide a comprehensive context for the presented findings.

Reviewer #3 (Remarks to the Author):

The revisions made by the authors have enhanced the manuscript, I suggest accept for publication in Nature Communication provided the additional questions below are addressed:

1. In the response letter, the authors test the maximum spectral resolution of the proposed spectrometer, <1nm resolution is an excellent result for practical application scenarios, but as mentioned in the letter, the authors applied narrow input spectra for training to let the algorithm “prepared for” the specific testing spectra (though most of the relevant articles used the same strategy). Can the authors comment on the way of testing the spectral resolution, and provide the exact resolution for narrowband spectra while maintaining the basic function of reconstructing the broadband spectra?

2. Schemes utilizing multi-layer film encoders can be categorized into two sections: those with an F-P cavity structure and those without. The distinction between the two lies in their spectral responses. Can the authors additionally discuss the merits of the plasmonic nanoparticles-in-cavity microfilter array compared to the multi-layer film array (without a strict F-P cavity structure) with broadband spectral response (e.g. ACS Photonics, 2022, 10(1): 225-233. and Light Sci. Appl. 2021, 10(1): 108)? Since the proposed filter array include multi-layer structure, I recommend add more analogous articles (e.g. Nature Photonics, 2023, 17(3): 218-223 and ACS Photonics, 2022, 10(1): 225-233.) into comparison and list as references.

Revisions according to the comments by Reviewer 1:

General comment: “I have read the revised version of the manuscript, along with the revised SI and authors' replies.

The authors have done a good job in thoroughly revising their work, including a comparative table with other similar techniques, and an additional experimental demonstration of a CMOS-image-sensor based portable spectrometer with a plasmonic microfilter array, demonstrating the usefulness of their scheme.

From my perspective, the authors have satisfactorily addressed my previous comments to them, and have accordingly 'reflected' those crucial amendments in the revised main manuscript and SI.

As such, I feel that this updated, amended and extended version of the present timely work now makes a sufficiently strong 'case' as to merit publication in NCOMMs - I recommend acceptance as is.”

Reply:

We thank the reviewer very much for the reviewer's positive comment and recommendation of acceptance.

Revisions according to the comments by Reviewer 2:

Comment 1: “It appears that the authors have not sufficiently provided experimental data to demonstrate the advantages or novelty of integrating plasmonic nanoparticles inside the cavities. It is recommended that they compare the Q factor and transmittance spectra of an FP cavity with and without plasmonic nanoparticles. This comparison will be crucial in showcasing the potential benefits of integrating plasmonic nanoparticles, such as improved Q factor and performance, which would justify their inclusion.”

Reply and revision:

Thank the reviewer very much for the reviewer's comment. We agree that Q factor is important in conventional spectrometers using a dispersive optics or tunable filters. The spectral resolving ability of those conventional spectrometers mainly rely on the bandwidth of spectral filters. A tunable spectral filter with narrow bandwidth, i.e., a high Q factor, contributes to a high resolution in those spectrometers.

In a computational spectrometer, however, the great computing power of microelectronic chips has greatly alleviated such a requirement on spectral filters. Instead of individually measuring different bands

of a spectrum, one may use a set of distinct filters to encode an input spectrum and then employ a reconstruction algorithm to computationally reconstruct the spectrum. Like that encoding capacity can be increased by either bit depth or the length of coding, the resolution of a computational spectrometer can be enhanced by either improving the Q factor of filters or increasing the number of spectrum-disparate filters. Indeed, it is this strategy that has liberated the design of spectrometers and inspired the breaking forth of many novel miniaturized spectrometers in recent years. As shown in **Fig. R1**, many kinds of spectral filters without a very narrow pass or reflection band can be utilized to encode input spectra for the development of miniature spectrometers.

Editorial Note: figure redacted

Figure R1. Spectra of photonic microfilters/nanowires used in previously reported miniature spectrometers. (a) Transmission spectra of colloidal quantum dot (CQD) filters for miniature spectrometer (J. Bao, et al., *Nature* **523**, 67-70, 2015). (b) Spectral responses of each constituent unit in a single-nanowire spectrometer (Z. Yang, et al., *Science* **365**, 1017-1020, 2019). (c) Transmission spectra of perovskite quantum dot (PQD) films for miniature spectrometer (Z. Zhu, et al., *Light: Science & Applications* **9**, 73, 2020).

Following this new design strategy, we integrate plasmonic nanoparticles inside the cavities to enhance the spectrometer's performance by mainly increasing the number of spectrum-disparate filters. As shown in **Fig. 4**, the integration of plasmonic Ag nanoparticles (AgNPs) in FP cavities can lead to a strong coupling between the localized surface plasmon resonance of AgNPs and FP microcavity and thereby efficiently increase the number of spectrum-disparate filters.

To show the advantages of the integration of AgNPs in such a plasmonic filter array more clearly, we compared the performance of two spectrometers constructed using only 144 pure FP microcavity filters (which were fabricated by using 36 groups of fabrication conditions without printing of AgNPs and repeated for 4 times) and that using all 1152 filters (which each group of FP microcavity is further repeated to print with 28 more groups of different-sized AgNPs, i.e., $36 \times 4 + 36 \times 28 = 1152$ filters in total), respectively. As shown in **Fig. R2**, with the integration of AgNPs, the peak-wavelength root-mean-square error (RMSE) and the FWHM of reconstructed spectra can be reduced from 0.189 and 0.747 nm to 0.034 nm and 0.651 nm, respectively. It clearly shows the advantages of the integration of plasmonic nanoparticles in improving the performance of computational spectrometers.

Figure R2. Performance comparison between two computational spectrometers constructed by using 144 pure FP microcavity filters and the plasmonic microfilter array with 1152 elements. (a) Peak-wavelength root-mean-square error (RMSE). (b) The FWHM of reconstructed spectra. The tests were conducted by using 0.54-nm-wide single-peak training spectra.

These results have nothing special and agree well with our expectations. Therefore, we revised a bit the manuscript to clarify these points as:

“Particularly, complementary metal oxide semiconductor (CMOS) or charge-coupled device (CCD) image sensor-based miniature spectrometers have been widely considered as the most viable candidate to resolve the tradeoff between miniaturizing size and retaining performance. They have not only millions and even tens of millions of photosensors, but also a very compact size and robust performance, which were widely tested in smartphones and lab-on-a-chip systems. Like that encoding capacity can be increased by the length of coding, the resolution of such CMOS/CCD image sensor-based computational spectrometers can be efficiently enhanced by increasing the number of spectrum-disparate filters. Therefore, it has recently come to the fore in the development of miniature spectrometers by devising novel optical microfilter arrays with a great many spectrum-disparate elements and spectrum reconstruction algorithms.”

(Line 58-67, Page 2)

“Our plasmonic nanoparticles-in-cavity microfilter array provides a near-perfect solution to this challenge. Two independent parameters, i.e., the size of AgNPs and the length of FP cavity, can be harnessed to tune the transmission spectra of microfilters. Compared to the scheme using pure FP cavity-based microfilters (i.e., without plasmonic nanoparticles)³⁶, it can efficiently increase the number of spectrum-disparate microfilters and thereby solve the challenge of ultrahigh precision nanofabrication of a large number of different cavity lengths within a micrometer-scale thin film. Compared to the scheme using pure plasmonic nanoparticles (i.e., without FP microcavities), it can greatly enhance the spectral tailoring ability and avoid the tough requirement on the precise engineering of individual plasmonic nanostructures for the fabrication of a large-scale spectrum-disparate plasmonic microfilter array.”

(Line 396-406, Page 13)

Comment 2: “The high resolution achieved by the authors may not be practically relevant, considering they used 2436 devices (filters) to achieve this level of resolution. Such a large number of devices would not be feasible for applications relying on CMOS sensor technology.”

Reply and revision:

Thanks for the reviewer's comment. In our experiments, we fabricated a plasmonic microfilter array with 1152 elements (microfilters), which has 2364 transmission spectral peaks (see Line 222 in Page 8 and Line 232 in Page 9), but not 2364 devices or filters.

On the other hand, CMOS sensor technology has been greatly advanced in both resolution and signal-to-noise ratio in recent years. For instance, the CMOS sensor used in our experiments, i.e., IMX178 SONY Back-illuminated Exmor R, has 6.44 million (i.e., 3096×2080) pixels, and each of its pixels can record 14 bits of data. Therefore, compared to the pixel number of CMOS sensors, the number of our filters is very small, which indicates that there is still ample room for future development of CMOS sensor-based miniature spectrometers for e.g., ultrahigh-resolution spectroscopy and spectral imaging applications.

We have revised our manuscript to provide the pixel number information of the CMOS sensor as:

“.....a monochrome camera (ASI178MM camera, Suzhou ZWO CO., LTD., China) with 14-bit CMOS image sensor with 3096×2080 pixels (Sony IMX178 sensor, Sony Group Corporation, Japan), and a commercial optical spectrometer (USB4000, Ocean Optics, USA) is used to build a setup for training and testing of the plasmonic microfilter array-based computational spectrometer, as shown in Figure S7.”

(Line 335-340, Page 11)

Comment 3: “The comparison table should incorporate more up-to-date results from state-of-the-art research to provide a comprehensive context for the presented findings.”

Reply and revision:

Thank you for the reviewer's valuable suggestion. We have added more up-to-date results^{R1-R8} to the comparison table:

Table S1. Comparison with recently reported computational spectrometers.

Publication information	Working mechanism	CMOS/CCD or other detectors	Operation range	Spectral resolution
Compact spectrometer based on a disordered photonic chip (Nature Photonics 2013)	Plasmonic nanohole	Photodetector array	1500 ~ 1525 nm	0.75 nm
A colloidal quantum dot spectrometer (Nature 2015)	Perovskite quantum dot	CCD sensor (Transmission mode)	390 ~ 690 nm	3.2 nm
Single-nanowire spectrometers (Science 2019)	Single compositionally engineered nanowire	Nanowire detector array	500 ~ 630 nm	7 ~ 8.5 nm
Single-shot on-chip spectral sensors based on photonic crystal slabs (Nature Communications 2019)	Photonic crystal slabs	CMOS sensor (Transmission mode)	550 ~ 750 nm	1 nm

Compact CMOS spectral sensor for the visible spectrum (Photonics Research 2019)	Photonic crystal slabs	CMOS sensor (Transmission mode)	400 ~ 700 nm	1 nm
Broadband perovskite quantum dot spectrometer beyond human visual resolution (Light: science & applications 2020)	Perovskite quantum dot	CCD sensor (Transmission mode)	250 ~ 1000 nm	1.6 nm
Neural network-based on-chip spectroscopy using a scalable Plasmonic Encoder (ACS Nano 2021)	Plasmonic nanohole	CMOS sensor (Transmission mode)	480 ~ 750 nm	<3.29 nm
A pearl spectrometer (Nano Lett. 2021)	Naturally pearl nanostructure	CCD sensor (Transmission mode)	450 ~ 700 nm	7.4 nm
Deeply learned broadband encoding stochastic hyperspectral imaging (Light: science & applications 2021)	Multilayer film filters (using SiO₂ and TiO₂ nanolayers)	CCD sensor (Transmission mode)	400 ~ 700 nm	5.2 nm
A wavelength-scale black phosphorus spectrometer (Nature Photonics 2021)	Tunable black-phosphorus detector	Black phosphorus detector	2000 ~ 9000 nm	420 nm
3D-printed miniature spectrometer for the visible range with a 100 × 100 μm ² footprint (Light: Advanced Manufacturing 2021)	3D-printed micro-optics	CMOS sensor (Transmission mode)	490 ~ 690 nm	9.2 nm
A single-dot perovskite spectrometer (Advanced Materials 2021)	LiCl-doped perovskite film	Perovskite detector	350 ~ 750 nm	5.3 nm
Mass production-enabled computational spectrometers based on multilayer thin films (Scientific Reports 2022)	Multilayer film filters (using SiO₂ and TiO₂ nanolayers)	CMOS sensor (Transmission mode)	500 ~ 849 nm	1 nm
Ultraspectral imaging based on metasurfaces with freeform shaped meta-atoms (Laser & Photonics Reviews 2022)	Metasurfaces with freeform shaped meta-atoms	CMOS sensor (Transmission mode)	460 ~ 740 nm	0.5 nm
Dynamic brain spectrum acquired by a real-time ultraspectral imaging chip with reconfigurable metasurfaces (Optica 2022)	Reconfigurable metasurfaces	CMOS sensor (Transmission mode)	450 ~ 750 nm	0.8 nm
Short-wave infrared chip-spectrometer by using laser direct-writing grayscale lithography (Advanced Optical Materials 2022)	Directly written Fabry-Pérot cavities	Photodetector array	900 ~ 1700 nm	2 nm
Miniaturized spectrometers with a tunable van der Waals junction (science 2022)	Tunable van der Waals junction	MoS₂/WSe₂ heterojunction detector	405 ~ 845 nm	3 nm
Deep learning-based miniaturized all-dielectric ultracompact film spectrometer (ACS Photonics 2023)	Five-layer film stacks (using SiO₂ and TiO₂ nanolayers)	CMOS sensor (Transmission mode)	400 ~ 700 nm	5 nm (16 filters) or 3nm (64 filters)
Video-rate hyperspectral camera based on a CMOS-compatible random array of Fabry-Pérot filters (Nature photonics 2023)	Fabry-Pérot filter array	CMOS sensor (Transmission mode)	400 ~ 700 nm	~20 nm
Imaging-based intelligent spectrometer on a plasmonic rainbow chip (Nature Communications 2023)	Plasmonic rainbow chip	CCD sensor (Reflection mode)	400 ~ 800 nm	2 nm
Folded digital meta-lenses for on-chip Spectrometer (Nano Letter 2023)	Folded digital meta-lenses	Photodetector	1530 ~ 1565 nm	0.14 nm
Inverse-designed linear coherent photonic networks for high-resolution	Integrated photonic chip	Photodetector	1515 ~ 1525 nm	0.1 nm

spectral reconstruction (ACS Photonics 2023)				
Integrated single-resonator spectrometer beyond the free-spectral-range limit (ACS Photonics 2023)	Single micro-ring resonator	Photodetector	1500 ~ 1600 nm	0.08 nm
Photon counting reconstructive spectrometer combining metasurfaces and superconducting nanowire single-photon detectors (Photonics Research 2023)	Metasurface array	Nanowire single-photon detectors	1500 ~ 1600 nm	2 nm
This work	Plasmonic nanoparticles-in-cavity microfilter array	CMOS sensor (Transmission mode)	395 ~ 725 nm	0.65 nm (0.49 nm for 580 ~ 680 nm)

We have revised our introduction for more up-to-date results and added them to the list of references:

“Therefore, many kinds of photonic technologies, such as thin-film optical filters¹², perovskite film¹³, single nanowire and superconducting nanowire^{14,15}, tunable van der Waals junction¹⁶, folded digital meta-lenses¹⁷, integrated photonic chips^{18,19}, and wavelength-selective photodetector^{20,21}, were demonstrated to build computational spectrometers toward chip-size integrated spectrometers.”

(Line 53-57, Page 2)

“Many kinds of microfilters using different techniques, such as quantum dots^{22,23}, photonic crystals²⁴⁻²⁷, plasmonic encoder²⁸, plasmonic rainbow chip²⁹, metamaterials^{6,11,15,30,31}, liquid crystal³², random structure^{33,34}, multi-layer film filters and interference filters^{12,35-40}, and 3D-printed micro-optics⁴¹, have been demonstrated to make computational spectrometers.”

(Line 67-71, Page 2)

- [R1] Zheng, J., Xiao, Y., Hu, M., Zhao, Y., Li, H., You, L., *et al.* Photon counting reconstructive spectrometer combining metasurfaces and superconducting nanowire single-photon detectors. *Photonics Research* 2023, **11**(2): 234-244.
- [R2] Yoon H.H., Fernandez H.A., Nigmatulin F., Cai W., Yang Z., Cui H., *et al.* Miniaturized spectrometers with a tunable van der Waals junction. *Science* 2022, **378**(6617): 296-299.
- [R3] Zhang Z, Liu Y, Wang Z, Zhang Y, Guo X, Xiao S, *et al.* Folded Digital Meta-Lenses for on-Chip Spectrometer. *Nano Letters* 2023, **23**(8): 3459-3466.
- [R4] Li Y, Zhang Z, Wang Y, Yu Y, Zhou X, Tsang HK, *et al.* Inverse-Designed Linear Coherent Photonic Networks for High-Resolution Spectral Reconstruction. *ACS Photonics* 2023, **10**(4): 1012-1018.
- [R5] Xu H, Qin Y, Hu G, Tsang HK. Integrated single-resonator spectrometer beyond the free-spectral-range limit. *ACS Photonics* 2023, **10**(3): 654-666.
- [R6] Zhang W, Song H, He X, Huang L, Zhang X, Zheng J, *et al.* Deeply learned broadband encoding stochastic hyperspectral imaging. *Light Sci. Appl.* 2021, **10**(1): 108.
- [R7] Wen J, Hao L, Gao C, Wang H, Mo K, Yuan W, *et al.* Deep Learning-Based Miniaturized All-Dielectric Ultracompact Film Spectrometer. *ACS Photonics* 2022, **10**(1): 225-233.
- [R8] Yako M, Yamaoka Y, Kiyohara T, Hosokawa C, Noda A, Tack K, *et al.* Video-rate hyperspectral camera based on a CMOS-compatible random array of Fabry-Pérot filters. *Nat. Photonics* 2023, **17**(3): 218-223.

Revisions according to the comments by Reviewer 3:

General comment: “The revisions made by the authors have enhanced the manuscript, I suggest accept for publication in Nature Communication provided the additional questions below are addressed”

Reply:

We greatly appreciate the reviewer's positive comment and suggestion on accepting for publication.

Comment 1: “In the response letter, the authors test the maximum spectral resolution of the proposed spectrometer, <1nm resolution is an excellent result for practical application scenarios, but as mentioned in the letter, the authors applied narrow input spectra for training to let the algorithm “prepared for” the specific testing spectra (though most of the relevant articles used the same strategy). Can the authors comment on the way of testing the spectral resolution, and provide the exact resolution for narrowband spectra while maintaining the basic function of reconstructing the broadband spectra?”

Reply and revision:

Thanks for the reviewer's comment. For quick and efficient reconstruction of broadband spectra, it is a common practice to use broadband input spectra to train the spectrometer, which can reduce the number of spectral peaks for spectrum reconstruction and therefore shorten testing time and lessen computational demand during both training and reconstruction processes^{R9}.

Nevertheless, according to the reviewer’s comment, we carried out more tests to use narrowband spectral peaks as training datasets and then measure and reconstruct both narrowband and broadband input spectra. Generally, broadband input spectra can also be well reconstructed. When the noises are relatively high, however, some small fake peaks originating from narrowband training spectra may appear on the envelop of reconstructed broadband spectra. To depress such a noise-sensitive problem, we further improved the spectrum reconstruction method by introducing additional step-size controls within the total variation (TV) and quadratic variation (QV) regularization terms, which can manage the correlation of weights for basis functions within a specific range for adapting to more diverse scenarios. Testing results indicate that the revised algorithm can remarkably enhance the performance on the reconstruction of broadband input spectra from narrowband training spectra. **Fig. R3** shows a testing result about the reconstruction of both narrowband (FWHM 1.37 nm) and broadband (FWHM 35.5 nm) input spectra using a series of narrowband single-peak training spectra. The cosine similarities of the two reconstructed spectra are 0.9985 and 0.9977, respectively.

Figure R3. Example of the reconstruction of narrowband (a) and broadband (b) input spectra using the same set of narrowband single-peak training spectra. Same training dataset but different hyperparameters are used to reconstruct these two input spectra. For the narrowband input spectrum (FWHM=1.37 nm), hyperparameter settings are $c_1 = 600$, $c_2 = 0.1$, $c_{3,1} = 0.1$, $c_{3,5} = 0.02$, $c_{3,9} = 0.05$, $c_{4,1} = 0.06$; For the broadband input spectrum (FWHM=35.5 nm), hyperparameter settings are $c_1 = 5.8 \times 10^5$, $c_2 = 1900$, $c_{3,1} = 9.86 \times 10^5$, $c_{3,5} = 1.59 \times 10^7$, $c_{3,9} = 0.1$, $c_{4,1} = 9430$. The other hyperparameters, i.e., $c_{3,p}$ ($p=2, 3, 4, 6, 7, 8$, or $p \geq 10$) and $c_{4,q}$ ($q \geq 2$), are all zero. The cosine similarities of the two reconstructed spectra are 0.9985 and 0.9977, respectively.

We have added **Fig. R3** to the Supplementary Information as **Fig. S9** and revised our manuscript to include the further improved spectrum reconstruction algorithm as:

“Therefore, the problem of finding w_j , where $j=1, \dots, n$, for the spectrum construction is transformed into solving the following optimization problem:

$$\begin{aligned} \text{Minimize } Loss(w_1, \dots, w_n) = & \sum_{k=1}^m \left(\tilde{y}_k - \sum_{j=1}^n w_j y_{j,k} \right)^2 + c_1 \sum_{j=1}^n |w_j| + c_2 \sum_{j=1}^n w_j^2 + \\ & \sum_{p=1}^{n-1} c_{3,p} \sum_{j=p+1}^n |w_j - w_{j-p}| + \sum_{q=1}^{n-1} c_{4,q} \sum_{j=q+1}^n (w_j - w_{j-q})^2, \end{aligned} \quad (5)”$$

(Line 283-285, Page 10)

“The hyperparameters $c_{3,p}$ ($p \geq 2$) control variants of TV, while $c_{4,q}$ ($q \geq 2$) control variants of QV. They introduce larger step sizes in TV and QV and should be set according to the spacing of the input light spectra in training. In practice, only a few of those terms are needed, and the other terms are allowed to be set to 0.

Notably, our machine learning algorithm is a generalization of various algorithms, such as least squares, LASSO, ridge regression, elastic net, fused LASSO, TV, and QV, and thus adapts to different practical cases by using different hyperparameters. When $c_1=c_2=c_{3,p}=c_{4,q}=0$ ($p \geq 1, q \geq 1$), the optimization reduces to least squares. When $c_2=c_{3,p}=c_{4,q}=0$ ($p \geq 1, q \geq 1$), the optimization becomes LASSO. When $c_1=c_{3,p}=c_{4,q}=0$ ($p \geq 1, q \geq 1$), the optimization is ridge regression. When $c_{3,p}=c_{4,q}=0$ ($p \geq 1, q \geq 1$), the optimization appears as an elastic net, which is the combination of LASSO and ridge regression⁴⁸. When $c_2=c_{3,p}=c_{4,q}=0$ ($p \geq 2, q \geq 1$), the optimization comes to be fused LASSO.

When $c_1=c_2=c_{3,p}=c_{4,q}=0$ ($p \geq 2, q \geq 1$), the optimization is TV. When $c_1=c_2=c_{3,p}=c_{4,q}=0$ ($p \geq 1, q \geq 2$), the optimization is QV.

(Line 312 - 324, Page 11)

“Another testing of the measurement of narrowband and broadband input spectra using the same set of narrowband single-peak training spectra is shown in Figure S10. These results show that the spectrometer can generally measure various input spectra with different profiles.”

(Line 386 -389, Page 13)

[R9] Tua, D., Liu, R., Yang, W., Zhou, L., Song, H., Ying, L., Gan, Q.. Imaging-based intelligent spectrometer on a plasmonic rainbow chip. *Nat. Commun.* 2023, **14**: 1902.

Comment 2: “Schemes utilizing multi-layer film encoders can be categorized into two sections: those with an F-P cavity structure and those without. The distinction between the two lies in their spectral responses. Can the authors additionally discuss the merits of the plasmonic nanoparticles-in-cavity microfilter array compared to the multi-layer film array (without a strict F-P cavity structure) with broadband spectral response (e.g. ACS Photonics, 2022, 10(1): 225-233. and Light Sci. Appl. 2021, 10(1): 108)? Since the proposed filter array include multi-layer structure, I recommend add more analogous articles (e.g. Nature Photonics, 2023, 17(3): 218-223 and ACS Photonics, 2022, 10(1): 225-233.) into comparison and list as reference.”

Reply and revision:

Thanks for the reviewer’s valuable comment. Compared with FP cavity filters, multi-layer film filters have more flexibility in tailoring spectral response. Compared with our proposed plasmonic nanoparticles-in-cavity microfilters, however, such a multi-layer film structure has less scalability in the fabrication of large-scale spectrum-disparate microfilter array. It was demonstrated that over ten steps of evaporation and patterning processes need to be applied to fabricate a multi-layer film filter array with 16 spectrum-disparate elements^{R6}. By contrast, our plasmonic microfilter array can be efficiently fabricated by using digital ultraviolet exposure technology. Microfilter array samples of over 1000 spectrum-disparate elements have been demonstrated to make high-resolution computational spectrometers in our experiments.

We have revised our manuscript to include a short comparison with a multi-layer film array as:

“Many kinds of microfilters using different techniques, such as quantum dots^{22,23}, photonic crystals²⁴⁻²⁷, plasmonic encoder²⁸, plasmonic rainbow chip²⁹, metamaterials^{6,11,15,30,31}, liquid crystal³², random structure^{33,34}, multi-layer film filters and interference filters^{12,35-40}, and 3D-printed micro-optics⁴¹, have been demonstrated to make computational spectrometers. Multi-layer film filters have good flexibility in tailoring optical spectral responses³⁷⁻³⁹. However, their fabrication processes using many steps of deposition/patterning are less scalable in the fabrication of large-scale distinct filter array. Nanophotonic methods that harness optical responses of nanostructures via material and geometric nanoengineering have become one of the most promising approaches. For instance, plasmonic 2D chirped gratings, in which the rainbow trapping effect is utilized to split wavelengths⁴², have been used to make a dual-functional computational spectrometer for simultaneous spectroscopic and polarimetric analysis.

However, the optical transmittance of such a metallic 2D chirped gratings is low and thus the computational spectrometer can work in reflection mode only, which may limit its further miniaturization for portable applications.”

(Line 67-80, Page 2-3)

Three more articles mentioned in these discussions have been added to the list of references as:

[R6] Zhang W, Song H, He X, Huang L, Zhang X, Zheng J, et al. Deeply learned broadband encoding stochastic hyperspectral imaging. *Light Sci. Appl.* 2021, **10**(1): 108.

[R7] Wen J, Hao L, Gao C, Wang H, Mo K, Yuan W, et al. Deep Learning-Based Miniaturized All-Dielectric Ultracompact Film Spectrometer. *ACS Photonics* 2022, **10**(1): 225-233.

[R8] Yako M, Yamaoka Y, Kiyohara T, Hosokawa C, Noda A, Tack K, et al. Video-rate hyperspectral camera based on a CMOS-compatible random array of Fabry–Pérot filters. *Nat. Photonics* 2023, **17**(3): 218-223.

Moreover, the experimental results about the performance testing of our computational spectrometer have also been further improved a little bit because of our recent upgrade of the light source of the spectrometer training/testing setup. We obtained a new xenon lamp with stronger and more stable light output in the visible light range (from 395 nm to 725 nm) and then used it to replace the halogen lamp in the previous setup. Therefore, we used a series of single-peak light spectra with narrower FWHM, i.e., changed from 0.65 nm to 0.54 nm, to train and test the spectrometer in the wavelength range from 395 nm to 725 nm. Consequently, the average value of measured FWHMs has been improved from 0.79 nm to 0.65 nm, as updated in **Figure 5c, 5b & 5c**. We have revised a bit the manuscript to update these results:

“In the experiments, a tunable optical monochromator with its broadband halogen tungsten light source (WDG15-Z, Beijing Optical Century Instrument Co., LTD., China), a Xenon lamp light source (BBZM-3, Anhui Bobei Lighting Electrical Appliance Factory, China),”

(Line 333-335, Page 11)

“In the first group of testing, a series of single-peak light spectra with the full-width at half-maximum (FWHM) of ~0.54 nm was used for training and testing of the spectrometer.”

(Line 359-361, Page 12)

“More meticulous testing of the spectrometer’s performance over the whole visible range was conducted by sweeping such single-peak light spectra over the wavelength from 395 nm to 725 nm with a step of 0.2 nm. Figure 5c shows the results of the root-mean-square error (RMSE) of peak wavelengths and the average FWHM of the measured spectra. It can be seen from Figure 5c(i) that the spectrometer can measure peak wavelengths accurately over the whole testing wavelength range (i.e., 395 ~ 725 nm) and its error is about 0.03 nm. The average value of measured FWHMs is 0.65 nm, which is close to the FWHM of the input spectrum, i.e., 0.54 nm.”

(Line 368-374, Page 12-13)

“With a machine-learning-based training process, an AI-empowered miniature spectrometer has been demonstrated to measure different input spectra at the average spectral resolution of 0.65 nm over the whole visible light range.”

(Line 492-495, Page 15)

“The spectrum-tunable light source used in the training and testing of the computational spectrometer was a tungsten halogen lamp plus a grating-based monochromator (WDG15-Z, Beijing Optical Century Instrument Co., LTD., China), and a xenon lamp light source (BBZM-3, Anhui Bobei Lighting Electrical Appliance Factory, China).”

(Page 3 of Supplementary Information)

Finally, we would like to thank the reviewers once again for their valuable comments, which have helped to further improve the quality of our manuscript.

REVIEWERS' COMMENTS

Reviewer #2 (Remarks to the Author):

I thank the response and the efforts that the authors have put into the manuscript focusing on plasmonic nanoparticles-in-cavity, which form the cornerstone of the work. While I appreciate the comparative analysis presented in Figure R2 between the two spectrometers, one with 144 FP cavities and the other with 1152 plasmonic filters, I have serious concerns regarding the fairness and relevance of this comparison.

To more effectively demonstrate the advantages of using plasmonic nanoparticles-in-cavity, a comparison with a closer parity in terms of the number of filters would be beneficial. For instance, contrasting the results from a spectrometer with 144 FP cavities against another with approximately 144 plasmonic filters would offer a more balanced and direct comparison.

Furthermore, to convincingly illustrate the key benefits of integrating plasmonic nanoparticles into the cavity, it is suggested a detailed comparative analysis focusing on the filtering effects. This could involve a side-by-side comparison of a pure FP cavity and a similar cavity augmented with plasmonic nanoparticles (with different sizes). Such a comparison would be instrumental in highlighting the distinct advantages and potentially unique characteristics introduced by the plasmonic nanoparticles.

Should the manuscript fail to substantiate these claims (true benefits of using plasmonics-in-cavity) effectively, its publication could be viewed as premature.

Reviewer #3 (Remarks to the Author):

The authors have made a thorough revision as suggested. I recommend acceptance to publish in Nature Communication with the revised version.

Reply to the comment by Reviewer 2:

Comment 1: “I thank the response and the efforts that the authors have put into the manuscript focusing on plasmonic nanoparticles-in-cavity, which form the cornerstone of the work. While I appreciate the comparative analysis presented in Figure R2 between the two spectrometers, one with 144 FP cavities and the other with 1152 plasmonic filters, I have serious concerns regarding the fairness and relevance of this comparison.

To more effectively demonstrate the advantages of using plasmonic nanoparticles-in-cavity, a comparison with a closer parity in terms of the number of filters would be beneficial. For instance, contrasting the results from a spectrometer with 144 FP cavities against another with approximately 144 plasmonic filters would offer a more balanced and direct comparison.

Furthermore, to convincingly illustrate the key benefits of integrating plasmonic nanoparticles into the cavity, it is suggested a detailed comparative analysis focusing on the filtering effects. This could involve a side-by-side comparison of a pure FP cavity and a similar cavity augmented with plasmonic nanoparticles (with different sizes). Such a comparison would be instrumental in highlighting the distinct advantages and potentially unique characteristics introduced by the plasmonic nanoparticles.

Should the manuscript fail to substantiate these claims (true benefits of using plasmonics-in-cavity) effectively, its publication could be viewed as premature.”

Reply:

Thanks for the reviewer’s comment. However, please kindly note that the key benefit of our plasmonic nanoparticles-in-cavity filters is to increase the scalability of FP cavity microfilters to achieve a large-scale spectrum-disparate microfilter array for a better spectrometer, but not to improve the performance of spectrometers by replacing FP cavity microfilters with the same amount of new plasmonic microfilters. We have expressly clarified it at several places in the manuscript, such as:

“The high scalability of such technological approaches may open a pathway for developing high-performance miniature optical spectrometers for extensive applications.”

(Line 18-19, in Abstract part)

“Compared to the scheme using pure FP cavity-based microfilters (i.e., without plasmonic nanoparticles), it can efficiently increase the number of spectrum-disparate microfilters and thereby solve the challenge of ultrahigh precision nanofabrication of a large number of different cavity

lengths within a micrometer-scale thin film.”

(Line 394-397, in Discussion part)

Indeed, as these 1152 plasmonic nanoparticles-in-cavity microfilters are directly extended from 144 pure FP cavities by direct printing of size-controlled AgNPs within 8 groups of same 144 FP cavities, the comparison between the results of 144 pure FP cavities and such 1152 plasmonic microfilters can show exactly the benefit of our scheme, i.e., to efficiently increase the number of spectrum-disparate microfilters to enhance the performances of miniature computational spectrometers. Compared to 144 pure FP cavities, the total thickness of 1152 microfilters increases only by dozens of nanometers, i.e., the diameters of AgNPs, but the performances of spectrometer have been dramatically enhanced in terms of spectral resolution and peak wavelength error (as demonstrated in the previous round of revision).

Nevertheless, we have tried a comparison of the performances between two spectrometers constructed by 144 pure FP microcavity filters and 144 plasmonic nanoparticles-in-cavity microfilters, respectively, as shown in Figure R1. It can be seen that the achieved performances of the two spectrometers are quite close: the peak-wavelength root-mean-square errors (RMSEs) of reconstructed spectra are reduced by using new plasmonic microfilters from 0.189 nm to 0.162 nm, while their average FWHMs of reconstructed spectra are reduced from 0.747 nm and 0.717 nm. Together with the previous results with 1152 plasmonic microfilters, they further verified that the efficient way to enhance the performances of miniature computational spectrometers is to increase the number of spectrum-disparate microfilters, but not to replace pure FP microcavity filter with plasmonic microfilters.

Figure R1. Performance comparison between two computational spectrometers constructed by using 144 pure FP microcavity filters and the plasmonic microfilter array with 144 elements. (a) Comparison of peak-wavelength root-mean-square error (RMSE). (b) Comparison of the average FWHMs of reconstructed spectra. The tests were conducted by using 0.54-nm-wide single-peak training spectra.

Regarding the side-by-side comparisons of a pure FP cavity and a plasmonic microfilter augmented with plasmonic nanoparticles (with different sizes), we have systematically discussed them in the manuscript and presented the numerical simulation and experimental results in Figures 2a & 2e and Figure 4d, respectively. These results revealed that the embedding of size-controlled AgNPs within FP cavities can efficiently alter the spectra of FP cavities via a strong coupling between the localized surface plasmon resonances of AgNPs and FP cavity modes and thus is a very promising pathway to create large-scale spectrum-disparate microfilter arrays for high-resolution broad-bandwidth miniature computational spectrometers.

Reply to the comment by Reviewer 3:

General comment: “The authors have made a thorough revision as suggested. I recommend acceptance to publish in Nature Communication with the revised version.”

Reply:

We thank the reviewer for his/her positive comment and recommendation of acceptance for publication.